# P32-specific CAR T cells with dual antitumor and antiangiogenic therapeutic potential in gliomas

Liat Rousso-Noori [1,12], Ignacio Mastandrea[1,12], Shauli Talmor[1], Tova Waks[2,3], Anat Globerson Levin[2], Maarja Haugas[4], Tambet Teesalu[4,5,6], Luis Alvarez-Vallina[7,8,9], Zelig Eshhar[2,3,10] & Dinorah Friedmann-Morvinski [1,11✉]

Glioblastoma is considered one of the most aggressive malignancies in adult and pediatric patients. Despite decades of research no curative treatment is available and it thus remains associated with a very dismal prognosis. Although recent pre-clinical and clinical studies have demonstrated the feasibility of chimeric antigen receptors (CAR) T cell immunotherapeutic approach in glioblastoma, tumor heterogeneity and antigen loss remain among one of the most important challenges to be addressed. In this study, we identify p32/gC1qR/HABP/C1qBP to be specifically expressed on the surface of glioma cells, making it a suitable tumor associated antigen for redirected CAR T cell therapy. We generate p32 CAR T cells and find them to recognize and specifically eliminate p32 expressing glioma cells and tumor derived endothelial cells in vitro and to control tumor growth in orthotopic syngeneic and xenograft mouse models. Thus, p32 CAR T cells may serve as a therapeutic option for glioblastoma patients.

[1] School of Neurobiology, Biochemistry and Biophysics, The George S. Wise Faculty of Life Sciences, Tel Aviv University, Tel Aviv, Israel. [2] Tel Aviv Sourasky Medical Center (TASMC), Tel Aviv, Israel. [3] Department of Immunology, Weizmann Institute of Science, Rehovot, Israel. [4] Laboratory of Cancer Biology, Institute of Biomedicine, Centre of Excellence for Translational Medicine, University of Tartu, Tartu, Estonia. [5] Cancer Research Center, Sanford Burnham Prebys Medical Discovery Institute, La Jolla, CA, USA. [6] Center for Nanomedicine and Department of Cell, Molecular and Developmental Biology, University of California Santa Barbara, Santa Barbara, CA, USA. [7] Cancer Immunotherapy Unit (UNICA), Department of Immunology, Hospital Universitario 12 de Octubre, Madrid, Spain. [8] Immuno-Oncology and Immunotherapy Group, Instituto de Investigación Sanitaria 12 de Octubre (imas12), Madrid, Spain. [9] Immunotherapy and Cell Engineering Laboratory, Department of Engineering, Aarhus University, Aarhus, Denmark. [10] Sackler Faculty of Medicine, Tel Aviv University, Tel Aviv, Israel. [11] Sagol School of Neuroscience, Tel Aviv University, Tel Aviv, Israel. [12]These authors contributed equally: Liat Rousso-Noori, Ignacio Mastandrea. ✉email: dino@tauex.tau.ac.il

Malignant gliomas are the most common primary brain tumors in the central nervous system, presenting highly infiltrative characteristics and dismal outcomes. Glioblastoma (GBM) is the most aggressive and lethal form among malignant gliomas with an average survival time of 12–18 months[1]. The current standard of care for newly diagnosed GBM patients consists of maximal surgical resection, radiotherapy, and concomitant chemotherapy (temozolomide). Unfortunately, this line of treatment has limited efficacy and the disease progresses or relapses. There is no effective treatment to offer to recurrent GBM patients.

Tumor immunotherapy has become the center of attention in the past decade, with adoptive cell transfer and checkpoint blockade having striking success in the clinics[2,3]. Chimeric antigen receptors (CARs) combine both antibody-like recognition and T cell activating function, endowing the engineered T cells with the capacity to recognize and kill cancer cells[4]. CAR T cell therapy targeting the CD19 receptor in patients with hematological malignancies ("liquid" cancer) has shown remarkable success[5–8], but initial attempts to use the same approach in treating "solid" tumors have encountered several challenges. One of the limitations in solid tumors is the lack of sufficient and specific targets, which can lead to CAR T cells with potential and dangerous "off tumor on target" toxicity. Three published clinical trials have been reported using CAR T therapy in GBM. The targeted antigens in these studies include epidermal growth factor receptor variant III (EGFRvIII)[9], human epidermal growth factor receptor 2 (HER2)[10], and interleukin receptor 13Rα2 (IL-13Rα2)[11]. Although all three studies reported evidence for the specific killing of tumor cells expressing these antigens, limited antitumor response has been observed in these clinical trials and patients eventually succumb to the disease.

We have previously reported the successful treatment of preclinical GBM models with a nanosystem targeted to tumor vasculature. This system consists of nanoparticles coated with a tumor-homing peptide, CGKRK, that specifically delivers its payload to tumor cells and tumor endothelial cells in GBM[12]. We identified p32/gC1qR/HABP/C1qBP to be the receptor for the CGKRK peptide, expressed in high levels on the surface of tumor cells and tumor-associated endothelial cells[13]. P32, also known as complement component 1, Q subcomponent-binding protein (C1QBP), is predominantly localized in the mitochondrial matrix, where it exerts its function in maintaining oxidative phosphorylation[14–18] and regulating the synthesis of mitochondrial-DNA-encoded genes[16,19].

In this study, we validate the expression of p32 in malignant gliomas and confirm its expression on the cell surface of tumor cells. We then focus our efforts to design a CAR targeting p32 positive glioma cells and provide proof-of-principle evidence that p32-CAR T cells are able to recognize and eliminate not only glioma cells but also tumor endothelial cells. We show that treatment with p32-CAR T cells reduces tumor vascularization and extends the overall survival of mice bearing gliomas. Our data suggest that p32 is a tumor-associated antigen (TAA) in gliomas and that CAR T cell immunotherapy against this target may be employed to achieve both antitumor and anti-angiogenic positive outcomes.

## Results

**p32 is expressed in murine and human glioma.** Others and we have identified p32 as the receptor for three tumor-homing peptides targeted nanoparticles: Lyp-1, CGKRK, and LinTT1[13,20,21]. p32 expression is significantly up-regulated in human cancers compared to their corresponding normal tissue[20], and although it is primarily expressed in the mitochondria[17,22], several studies reported the expression of p32 on the surface of malignant

cells[23,24]. We sought to determine the expression profile of p32 in murine and human gliomas. Using Rembrandt datasets we first confirmed upregulation of *p32* mRNA in low and high-grade gliomas compared to non-tumor tissue and in all three molecular subtypes of GBM (Supplementary Fig. 1a, b). We exploited a cohort of paired primary and recurrent GBM samples and found higher *p32* mRNA levels in recurrent GBM (Supplementary Fig. 1c). Kaplan–Meier survival plot showed that increased expression of *p32* in malignant gliomas is associated with worst prognosis with decreased overall survival rates (Supplementary Fig. 1d). Next, human glioma specimens were stained with p32 Ab showing significant enhanced expression with tumor grade and compared to normal brain tissue (Fig. 1a). Similar results were previously observed when p32 expression was assessed using a brain tumor tissue array, showing significant upregulation of p32 in higher grade gliomas compared to normal brain tissue[15]. We validated p32 protein expression by western blot analysis in murine GBM cells, patient-derived GBM83 glioma stem cell (GSC) and human established glioma cells (Supplementary Fig. 1e) and by confocal microscopy in a syngeneic and PDX GBM mouse model, confirming specific expression in tumors but not in normal brain tissue (Fig. 1b). Finally, we examined the expression of p32 on the surface of several murine gliomas derived cells established from our lentiviral-induced adult and pediatric glioma mouse model (005, AFFR53, and O1), as well as on human established cell lines (U87, U118, U178, and U251), and patient-derived glioma stem cells (PD-GSCs) by flow cytometry analysis (Fig. 1b and Supplementary Fig. 2a, b). Among these PD-GSCs we have representatives of both proneural (GBM1079 and GBM1051) and mesenchymal (GBM83, GBM1005, GBM1027) GBM molecular subtypes (Fig. 1b and Supplementary Fig. 2a). All glioma cells stained positive for p32 expression on the cell surface using the same anti-p32 mAb (see the "Methods" section), while human primary cells evaluated were negative (Fig. 1c, d and Supplementary Fig. 3). Besides, we examined the intracytoplasmic and surface expression of p32 in our murine glioma-derived cells and human glioma cells in comparison with primary cortical astrocytes and fibroblasts, and further confirmed that surface expression is restricted to tumor cells (Fig. 1d). Altogether these findings suggest p32 may serve as a TAA in low- and high-grade gliomas.

**Generation and characterization of murine and human p32-specific CAR T cells.** To explore whether the expression of p32 on the surface of glioma cells could serve as an alternative TAA for adoptive cell immunotherapy of brain tumors, a p32-specific 2nd generation CAR construct was designed (Fig. 2a). To generate p32-CAR we cloned a single-chain variable fragment (scFv), clone 2.15, which is directed against the same C1q binding domain of p32 as mAb clone 60.11 used in all our staining experiments described in the previous section. P32 is a highly conserved protein and both the antibody and the scFv clone 2.15 recognize human and murine p32[25]. The scFv is followed by transmembrane and cytoplasmic CD28 and intracellular FcγR domains capable of activating both murine and human T cells[26–28]. We added a FLAG-tag at the N-term of the scFv for easy validation of expression of the CAR construct on the surface of the transduced T cells and a mcherry fluorescent reporter (separated from the CAR cassette by a P2A skip sequence) to assess transduction efficiency (Fig. 2a). The transduction efficiency of both murine and human activated T cells using the same CAR construct was on average 30–40% (Fig. 2b, c). The phenotypic analysis of CD4 and CD8 subpopulations showed no major changes in their percentages following transduction with the CAR expressing retroviral particles and both CD4 and CD8 subpopulations expressed the p32 CAR (Fig. 2d, e). T cell activation levels were indistinguishable between controls and CAR⁺

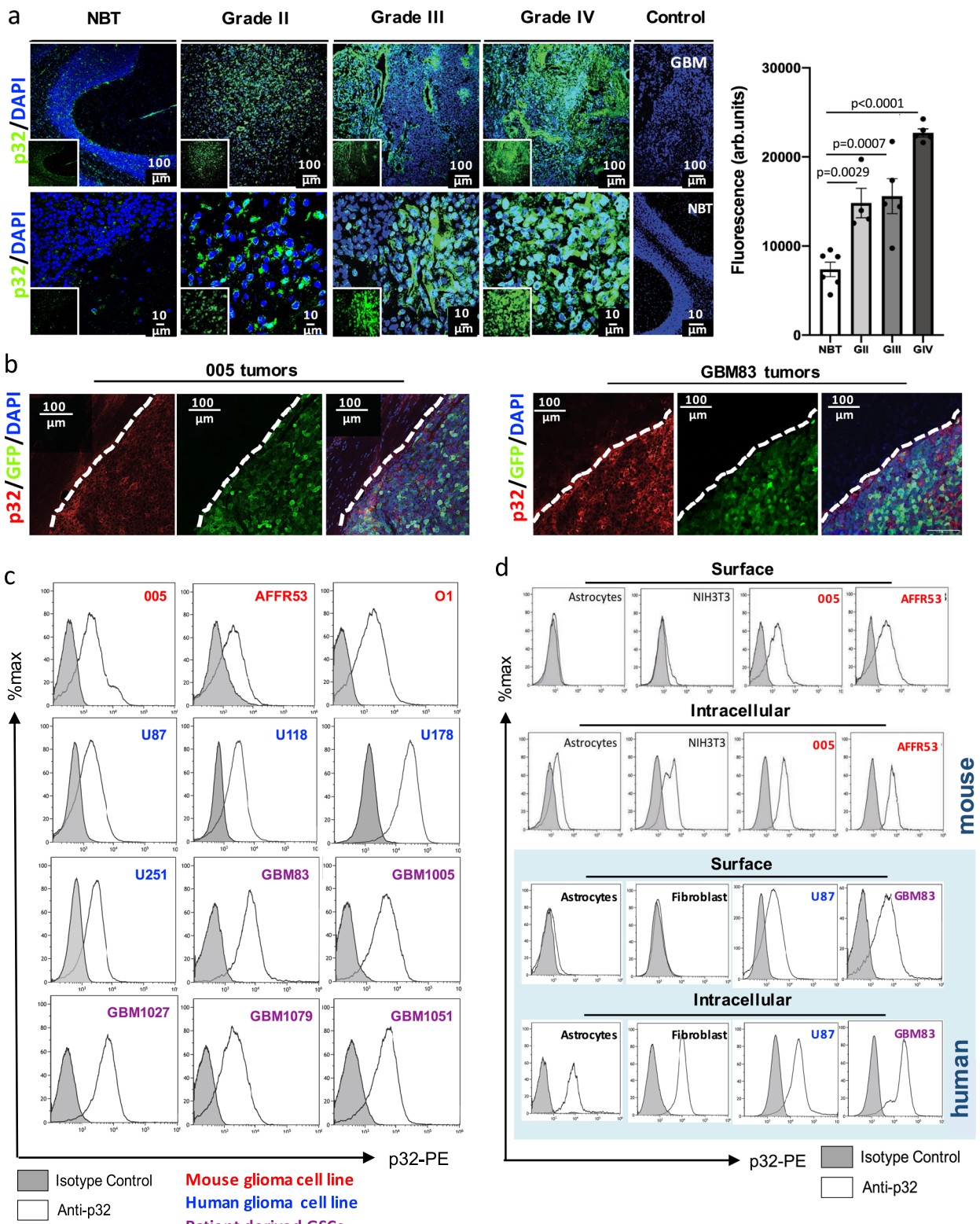

T cells (Fig. 2f, g), and no differences were observed in the expression of representative exhaustion markers (Fig. 2h, i), suggesting that the introduction of the CARs into T cells did not result in tonic signaling. Phenotypic analysis of human lymphocytes showed that p32 CAR+ T cells contained central-memory, effector-memory, and T stem cell memory, without significant difference between untransduced and CAR+ expressing T cells (Fig. 2j).

**Functional evaluation of p32 CAR T cells in vitro.** Next, we evaluated the in vitro anti-tumor effect of murine and human p32 CAR T cells. To evaluate the functionality of p32 murine CAR T cells we used two different p32+ tumor-derived cell lines, one maintained in a differentiated state, AFFR53, and another line, 005 that we have previously characterized as GSC and that forms typical neurospheres (also termed "tumorspheres")[29] (Supplementary

**Fig. 1 Analysis of p32 expression levels in murine and human glioma samples. a** Confocal microscopy analysis of normal brain tissue (NBT), grade II diffuse astrocytoma, grade III anaplastic oligodendroglioma, and grade IV glioblastoma. Sections were stained with rabbit anti-p32 antibody. Negative control (Control) sections of GBM were incubated only with secondary Ab goat-anti-rabbit AlexaFluor488. The graph represents quantification of fluorescence intensity of p32 signal in normal brain and tumor sections from patients. Data represent mean ± SEM. Each dot represents the average of three images per sample. $N = 6$ (NBT), $N = 4$ (GII = Grade II), $N = 5$ (GIII = Grade III), and $N = 5$ (GIV = Grade IV) human samples per group. Statistical significance was determined using one-way ANOVA test with multiple comparisons adjustment. **b** Expression of p32 in murine 005 and patient-derived GBM83 xenograft tumors is confined just to the tumor area (GFP + area). Representative image of three independent mice per model. **c** p32 cell surface expression analyzed by FACS in murine glioma cells (005, AFFR53, O1), human cell lines (U87, U118, U178, U251) and patient-derived mesenchymal (GBM83, GBM1005, GBM1027) and proneural (GBM1079, GBM1051) GSCs. **d** FACS analysis of surface and intracellular expression levels of p32 in primary murine astrocytes, NIH3T3 fibroblasts and murine 005 and AFFR53 glioma cells (upper panels). Same analysis was performed with primary human astrocytes and skin fibroblasts, U87 and GBM83 human glioma cells (lower panels, light blue background). All histograms shown are representative of independent stainings for each cell line, $N = 3$ per cell line. All source data are provided as a Source Data file.

Fig. 4a, b and Supplementary Table 1). To further examine the specificity of the p32 mCAR T cells we knockdown the expression of p32 in these cells (Supplementary Fig. 4c, d). All the indicated tumor cell lines were co-cultured with control anti-TNP SP6 mCAR T[26] or p32 mCAR T cells at different effector T cell to tumor cell (E: T) ratios. As shown in Fig. 3a redirected p32 mCAR T cells effectively and specifically killed p32+ expressing glioma cells, while no effect was observed when co-cultured with p32KD cells. Control irrelevant SP6 mCAR T showed no cytotoxic activity when co-culture with all glioma cells. Specific cytotoxic activity and CAR+ T cell expansion were further confirmed by co-culture of SP6 and p32 mCAR T cells with GFP+ glioma cells and analyzed by flow cytometry (Supplementary Fig. 5a). T cell proliferation in response to p32+ glioma target cells was assessed by CellTracer™ Violet dilution assay (Fig. 3b). Next, we determined the tumor-specific recognition of p32-expressing glioma cells by both intracytoplasmic IFN-γ production (FACS analysis, Fig. 3c) and secretion to the culture media (Fig. 3d).

Similar line of experiments was conducted with transduced p32 human CAR T cells co-cultured with either human U87 glioma cells (maintained in the presence of serum; differentiated state) or GBM83 patient-derived GSCs[30]. Untransduced (UT) T cells were used as controls in all the experiments with human glioma cells[28,31]. While p32 hCAR T cells were able to exert their cytotoxic effect (Fig. 3e and Supplementary Fig. 5b), proliferate (Fig. 3f) and secrete IFN-γ (Fig. 3g) when co-culture with glioma cells, UT T cells showed little to no response. These results demonstrate that both murine and human p32 CAR T cells recognize p32+ glioma target cells specifically and are able to kill only p32-expressing glioma cells.

**P32 CAR T cells show antitumor activity in both syngeneic and xenograft models.** To evaluate the antitumor effect of p32 mCAR T cells in vivo in a syngeneic immunocompetent model, we transplanted 005 cells into the brain of C57BL/6 mice. Nine days later, mice were preconditioned with a single dose of 200 mg/kg of cyclophosphamide intraperitoneally, to create a lymphodepleted environment to favor the adoptive transfer of the engineered lymphocytes[28]. On day 10, mice were infused with $1 \times 10^7$ total T cells, containing $\sim 4 \times 10^6$ p32-specific mCAR+ T or Sp6-specific mCAR+ T cells (control group) intravenously ((i.v.) tail vein injection). A second dose of $1 \times 10^7$ ($4 \times 10^6$ mCAR+) T cells per mouse was infused seven days later. The results showed that mice receiving p32 mCAR T cells survived longer than the control group (median survival of 71 vs. 43 days, $p = 0.0413$, respectively) and 30% of the treated mice remained tumor free until the end of the experiment (Fig. 4a). No toxic or adverse effects were observed in the mice infused with p32 mCAR T cells (Supplementary Fig. 6a), suggesting specificity and safety of these CAR T cells in vivo. To further evaluate the potential toxicity of

p32 mCAR T cells, C57BL/6 J non-tumor-bearing mice were preconditioned and 24 h later were infused i.v. with either SP6 or p32 mCAR T cells. All mice were euthanized 2 weeks later to assess organ toxicity. No signs of evident lesions or inflammation were observed in the examined tissue sections (Supplementary Fig. 6b). Furthermore, lung tissue from infused mice were collected, dissociated and analyzed for CAR T proliferation and activation. Once again, no difference was observed between the irrelevant CAR and p32-specific CAR T cells (Supplementary Fig. 6c), indicating that off-target effect is not observed as the activation state of the injected CAR T cells is unchanged in healthy tissues such as the lungs.

Next, we tested the human transduced CAR T cells in two human orthotopic xenograft models. First, we used U87 cells transplanted in the brain of NUDE mice, a well-established preclinical and reproducible GBM xenograft model as proof of concept[32]. Ten days after intracranial injection of U87 glioma cells, a single dose of p32 hCAR T cells or UT T cells were infused intratumorally. Kaplan–Meier survival curve shows significantly improved overall survival results for mice treated with p32 hCAR T cells compared to the control group (Fig. 4b). For the next set of experiments we decided to use a more aggressive model of GBM, this time with patient-derived glioma stem cells (GBM83), derived from a mesenchymal aggressive tumor[30,33]. First, GBM83 cells were transduced with firefly luciferase, allowing us to track tumor growth using in vivo bioluminescence imaging, and then transplanted orthotopically in NUDE mice. The latency of GBM83 tumors is on average 20 days so three days after injection of the tumor cells (the presence of a lesion was confirmed by luciferase positive signal) mice received one dose ($1 \times 10^7$ cells) of UT or p32 hCAR T cells (containing $\sim 4 \times 10^6$ hCAR+ T cells) intratumorally and another dose intraventricularly ($2.5 \times 10^6$ total T cells, $\sim 1 \times 10^6$ CAR+) (Fig. 4c). The latest route of injection was selected based on the positive and improved results obtained in pre-clinical and clinical trials with CAR T cells in GBM[11]. While UT control mice showed continuous tumor growth and rapidly succumb to the disease, mice treated with p32 hCAR T cells showed prolonged survival (Fig. 4c). No signs of toxicity or adverse effects were observed in the treated mice (Supplementary Fig. 6d). Comparison of bioluminescence imaging data revealed a significant difference between days 6 and 17 of treatment between UT and p32 hCAR T treated group (Fig. 4d, e). One indication of on-target effect is the decrease of antigen expression in the remaining tumor after treatment. To that end, we analyzed at endpoint the resulting tumors after treatment with p32 hCAR T and UT cells, and compared their p32 expression by immunofluorescent confocal microscopy (Fig. 4f) and flow cytometry analysis (Fig. 4g). We found that p32 expression levels decreased in p32 hCAR T treated mice compared to UT control mice. Next, we sought to analyze the expression of PD−1 on gated CD3+ infiltrating T cells from both UT and p32 hCAR T treated tumors, and found that in all treated

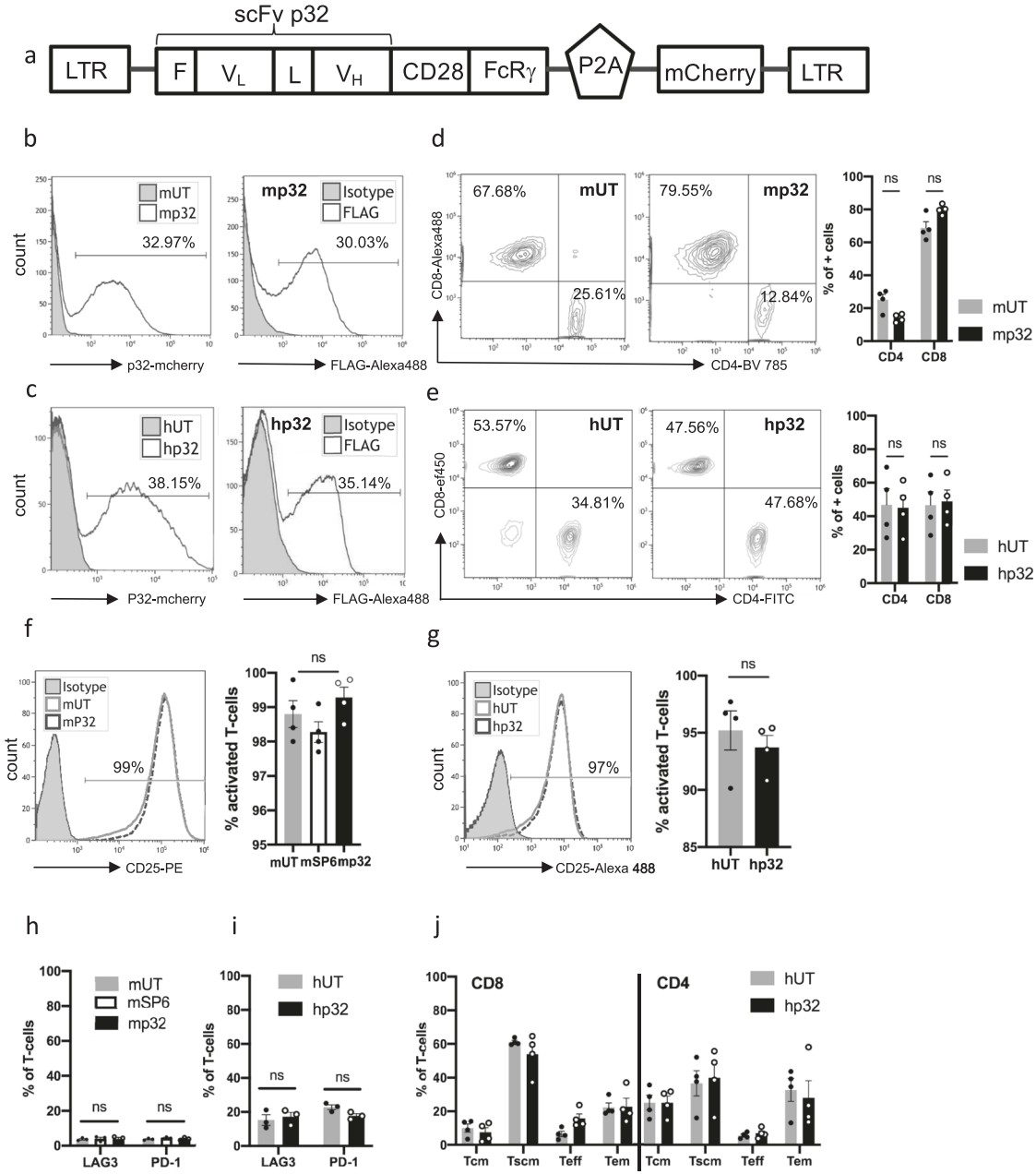

**Fig. 2 CAR design and human and murine T Cell transduction. a** Schematic representation of the retroviral vector expressing the p32 CAR (F = FLAG, L = Linker, LTR = long terminal repeat) and its transduction efficiency in mouse **b** and human **c** T cells. The transduction efficiency was measured by mcherry positive cells and by FLAG cell surface expression. Untransduced T cells were used as control (mUT = mouse and hUT = human). Histograms are representative of three independent experiments. **d, e** Frequency of CD4+/ CD8+ T cells 3 days post transduction showing no significant (ns) variation after CAR transduction of mouse (**d**) and human (**e**) T cells. Graphs on the right panel represent the quantification of $N = 4$ mice and $N = 4$ human donors, respectively. **f, g** Activation marker expression 2 days after transduction of mouse (**f**) and human (**g**) lymphocytes. Graphs on the right panel represent the quantification of $N = 4$ mice and $N = 4$ human donors, respectively. **h, i** Quantification of representative exhaustion marker expression 4 days (**h**, mouse, $N = 3$) and 10 days (**i**, human, $N = 3$ donors) after initial activation. **j** Phenotypic analysis of human UT (hUT) and CAR+(hp32) T cells at 10 days post transduction showing the frequency of central memory T cells (Tcm, CD45RA−CCR7+), stem central memory T cells (Tscm, CD45RA + CCR7+), effector T cells (Teff, CD45RA + CCR7−) and effector memory T cells (Tem, CD45RA−CCR7−) in CD8+ and CD4+ T cells; $N = 4$ donors. Each dot in the graphs represents a donor's average of three independent experiments and final data is presented as mean ± SEM. Statistical significance was determined using an unpaired, two-sided $t$ test when comparing between two groups (**d, e, g, i, j**), and multiple comparisons ANOVA test when comparing more than two groups (**f, h**). Source data are provided as a Source Data file.

samples PD−1 was highly expressed (Fig. 4h). Based on these results and previously published work evaluating patients treated with CAR T[9] we can only speculate that T cells in the tumors become exhausted and are probably affected by the glioma immunosuppressive microenvironment.

**Antiangiogenic effect of p32 CAR T cells**. When we first identified p32 as the receptor of the CGKRK homing-peptide nano-system, we not only found it expressed on glioma cells, like the 005 GSCs, but also in tumor-derived endothelial cells (TDEC) and tumor vasculature[12]. Under hypoxic conditions, 005 GSCs

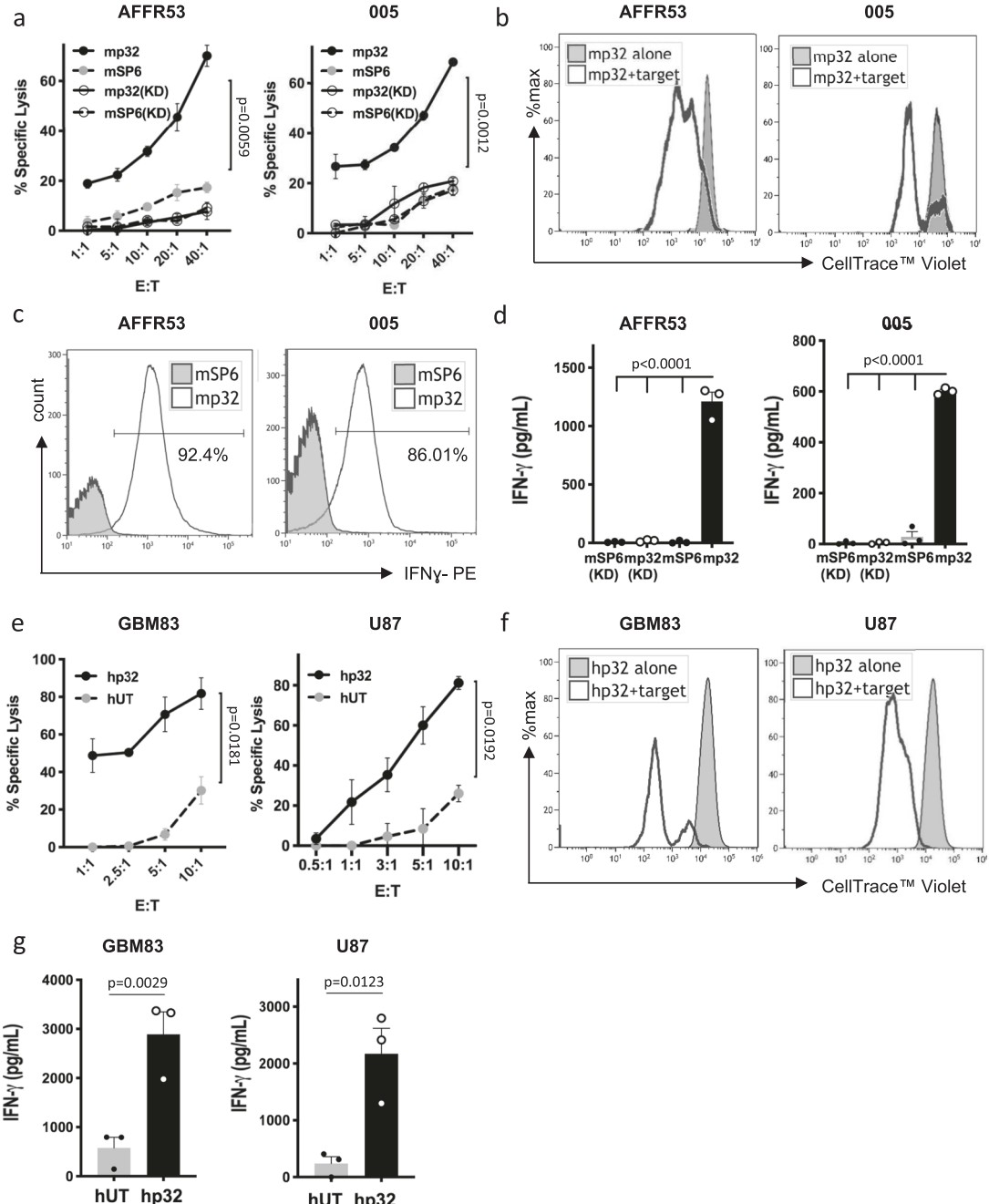

**Fig. 3 Murine and human p32 CAR T cells specifically target p32 expressing glioma cells. a** Specific cytotoxicity towards p32-positive and the corresponding p32 KD cells (knockdown; transduced with lentivirus expressing shRNA targeting p32) was measured by LDH activity in culture media. Target cells were co-cultured with either a non-relevant (mSP6) or p32 murine CAR T (mp32) cells for 6 h at the indicated E:T ratios. Data represent mean ± SEM. $N = 3$. One-way ANOVA, Tukey's multiple comparisons test. $P$ value shown in graph corresponds to the comparison of the mp32 vs mSP6. $P < 0.004$ for the remaining comparisons. **b** CellTracer™ Violet dilution assay of labeled p32 CAR T murine lymphocytes co-cultured with the indicated glioma cells for 3 days (E:T = 3:1). Labeled unstimulated p32 CAR T cells alone were used as control. Representative histogram is shown out of four independent experiments. **c, d** Transduced murine lymphocytes were incubated with the target cells (E:T = 2:1) for 24 h, and IFN-γ production was measured by FACS intracellular staining (**c**) and by ELISA (**d**). $N = 3$, representative histogram is shown. Dots in graph (**d**) show average for each independent experiment. Data are shown as mean ± SEM. Statistical significance was determined using one-way ANOVA test with multiple comparisons adjustment. **e** Human target glioma cells were co-cultured with control untransduced T cells (hUT) or CAR-p32 T cells (hp32) at the indicated E:T ratios. Eighteen hours later, the cytotoxic action of the human CAR T cells was measured by quantifying luciferase activity in tumor target cells. Data represent mean ± SEM. $N = 5$ for GBM and $N = 4$ for U87 independent experiments, two different human donors were used. Unpaired $t$ test with Welch's correction was used for statistical analysis. Two-tailed $P$ value is shown. **f** CellTracer™ Violet dilution assay (representative histograms from three independent experiments with two different donors) and **g** IFN-γ secretion (ELISA; $N = 3$ independent experiments with two different donors) were also assessed using hUT and hp32 T cells following co-culture with the indicated human glioma cells. Data are shown as mean ± SEM. Unpaired $t$ test was used. Two-tailed $P$ value is shown. All source data are provided as a Source Data file.

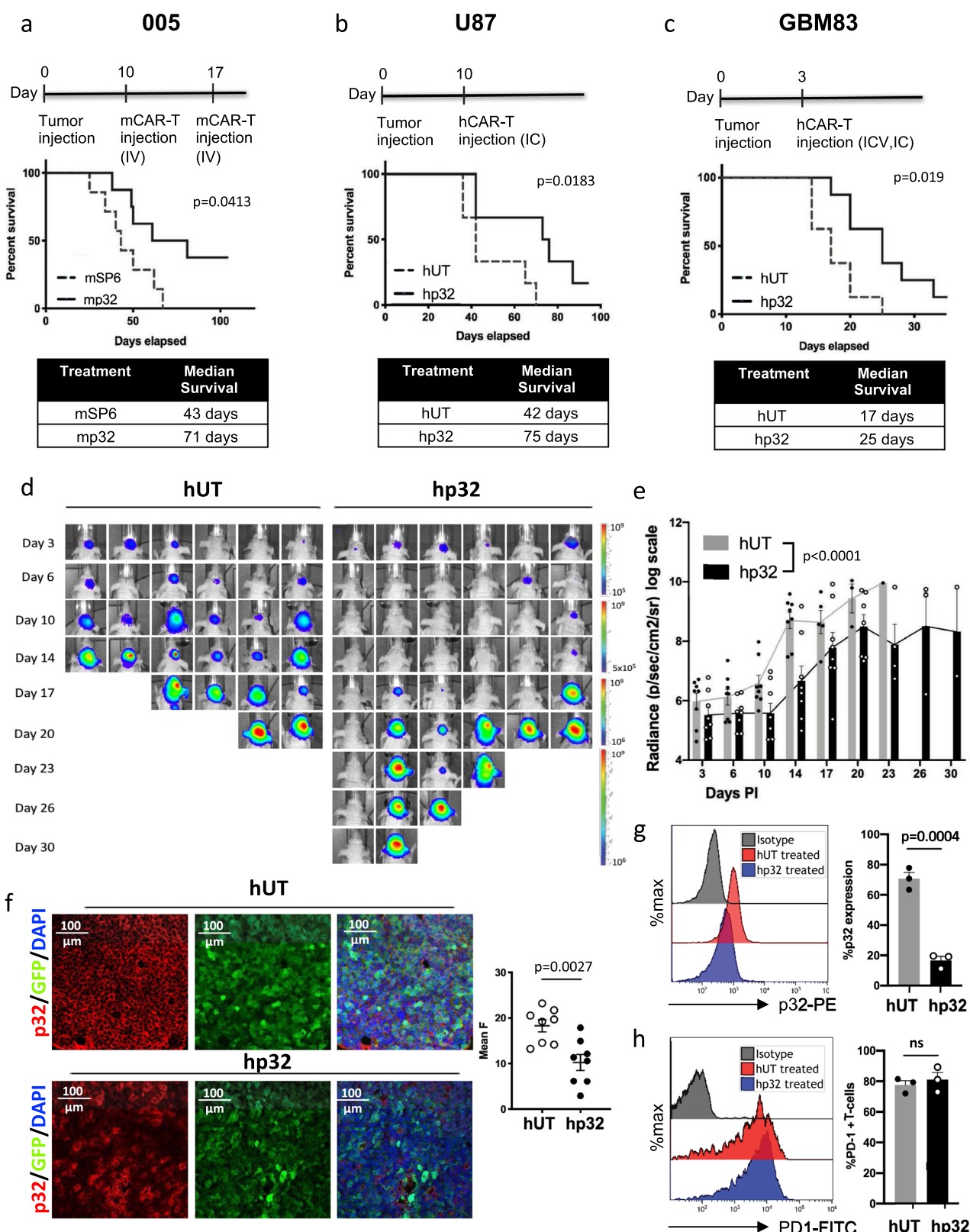

have the capacity to transdifferentiate and give rise to TDEC both in vitro (Fig. 5a) and in vivo[34,35]. Confocal microscopy (Fig. 5a) and FACS analysis[12] confirmed the expression of p32 in 005 cells cultured in EGM2 endothelial media supplemented with deferoxamine (DFO), an iron chelator that mimics hypoxic conditions by blocking proline hydroxylase[36,37]. Next, we co-culture p32 or SP6 mCAR T cells with 005 TDEC at different effector to target ratios and as shown in Fig. 5b p32 mCAR T cells effectively killed p32 expressing TDEC. The cytolytic activity of p32 mCAR T cells was corroborated by IFN-γ release in the culture supernatant (Fig. 5c). Interestingly, when we analyzed tumor sections from our previous in vivo experiments by confocal microscopy, we observed that while tumors in control groups infused with UT or control SP6 CAR T cells were highly vascularized and stained

**Fig. 4 Antitumor response of p32 CAR T cells against gliomas in vivo. a** Groups of 6–8 weeks C57BL/6J mice ($N = 8$) were transplanted with $3 \times 10^5$ 005 GSC, and then adoptively transferred intravenously (IV) with either mCAR-SP6 (mSp6) or mCAR-p32 (mp32) T cells ($1 \times 10^7$ total T cells) on days 10 and 17 post tumor implantation. **b** Groups of 6–8 weeks Nude mice ($N = 6$) were intracranially (IC) inoculated with $3 \times 10^5$ U87 cells, and then adoptively transferred with either human UT or CAR-p32 (hp32) T cells ($1 \times 10^7$ total) on day 10 IC. **c** Groups of 6–8 weeks NUDE mice ($N = 8$) were IC inoculated with $3 \times 10^4$ GBM83 cells, and then adoptively transferred with either hUT or hp32 T cells ($1 \times 10^7$ total) on day 3 IC and $2.5 \times 10^6$ total T cells intracerebroventricularly (ICV). Two-sided Mantel-Cox (log-rank) test was used to address differences in median survival in (**a–c**). **d, e** Representative bioluminescence images (**d**) and bioluminescence kinetics (**e**) of GBM83-luciferase-expressing cells engrafted in Nude mice. Data represent mean ± SEM, each dot represents one mouse, $N = 8$. Two-way ANOVA was used to analyze overall differences in both groups during the course of the experiment. **f, g** Analysis of p32 expression in GBM83 tumor-bearing mice following CAR T treatment. **f** Tumor sections (40 μm) were stained with p32 antibody and analyzed by confocal microscope (GFP = tumor area; DAPI = nuclei staining). Quantification was assessed by Fiji/ImageJ. Each dot shown in the graph represents average of three measurements done per slide. A total of eight slides originated in three different mice were stained per group. Unpaired $t$ test was used. Two-tailed $P$ value is shown. **g, h** Representative GBM83 tumors from each group were dissociated and analyzed by FACS for p32 surface expression on GFP+ tumor cells (**g**) and PD-1 expression of tumor infiltrated T lymphocytes (gated on CD3 + fraction) (**h**). Quantification of the results is summarized in the bars graph. Each dot represents the average value for one specific mouse ($N = 3$). Results are expressed as median ± SEM. For **g, h** unpaired $t$ test was used, two-tailed $P$ value is shown. All source data are available in the Source Data file.

positive for the endothelial marker vWF, the p32 CAR T cell treated mice showed significantly less blood vessels in the tumors (Fig. 5e and quantification by ImageJ in Fig. 5e). The same trend was observed when GBM83 tumors infused with either UT or p32 CAR T cells were dissociated at endpoint and analyzed by flow cytometry for CD31$^+$ staining (Supplementary Fig. 7). To further confirm that this anti-angiogenic effect is related to the expression of p32 on TDEC and tumor vasculature, we engineered GBM83 cells to overexpress ErbB2 (Supplementary Fig. 8a). Next we transduce T cells with an ErbB-2-specific CAR (hN29)[28] (Supplementary Fig. 8b) and evaluated the in vitro functionality of hN29 CAR T on ErbB-2-GBM83 cells (Supplementary Fig. 8c–e). Finally, we infused either UT or hN29 CAR T (same dose and location as with hp32 CAR T experiments) to ErbB-2-GBM83 injected mice and followed tumor growth and survival (Supplementary Fig. 8f–h). While we observed an effect on tumor growth and survival with the hN29 CAR T cell treatment, no difference in tumor vasculature was obtained (Supplementary Fig. 8i–k). Altogether, our results suggest that p32 CAR T cells are able not only to recognize tumor cells but also tumor-associated endothelial cells, suggesting a possible antiangiogenic effect of p32 CAR T cells by targeting the tumor vasculature.

## Discussion

Despite ongoing success in CD19$^+$ B cell hematologic malignancies with CAR T cell therapy, progress in the solid tumor arena still has to face many obstacles. Among the known list of challenges is the identification of a suitable neoantigen or TAA to re-direct the engineered CAR T cells. GBM shares a high level of biological complexity with other solid tumors, with an emphasis on molecular and cellular heterogeneity, the latest reflected by a spectrum of dynamic cell states[38]. Mono-specific CAR T cell therapy demonstrated feasibility and safety when tested in GBM, but eventually resulted in the outgrowth of tumors lacking the expression of the single antigen targeted[9]. Therefore, the identification of an array of glioma-associated antigens is critical for the strategic success of CAR T therapy in highly heterogeneous solid tumors like GBM.

We found in previous work that p32 is the binding partner of CGKRK and LinTT1 tumor-homing peptides-coated nanoparticles in GBM[12,21]. In this study, we confirmed and validated the upregulation and surface expression of p32 in glioma cells and associated endothelial cells. Most importantly, we could not detect p32 expression in normal brain tissue or healthy tissue from other organs.

Higher expression of p32 has been reported in several types of tumors including melanoma, colon, ovarian, endometrial, prostate, and breast, suggesting a potential role in tumorigenesis. Indeed,

genetic knockdown of p32 shifted the metabolism of tumor cells from oxidative phosphorylation to glycolysis and resulted in reduced tumor formation in vivo[16]. In gliomas, p32 expression was found to be significantly correlated with the expression of c-myc, and to be involved in the reprogramming of glutamine metabolism in these tumors[15]. Silencing p32 resulted in impaired cell proliferation in vitro and had an anti-tumor effect in vivo[15]. In line of this evidence, mitochondrial localized p32 has been recently evaluated as a therapeutic target in gliomas using a small molecule inhibitor[39]. While mitochondrial p32 functional role has been studied, its translocation to the cell surface is not completely understood. p32 is also known as hyaluronic acid binding protein (HABP)[40]. Hyaluronic acid (HA) is an important component of the brain extracellular matrix (ECM) and is one of the major components of GBM-ECM. In addition to anatomical and physical aspects, specific ECM components in GBM, such as Tenascin C, fibronectin, and hyaluronan were shown to contribute in different aspects of tumor biology (e.g., invasiveness, proliferation, angiogenesis)[35]. Based on a recently reported study[41], we speculate that increased levels of HA may disrupt the normal localization of p32 in the mitochondria and induce its translocation to the cell surface. P32 expressed on the cell surface of tumor cells can interact with HA and this interaction may contribute to GBM highly invasive and proliferative capacity.

In the present study, we took advantage of the selective expression of p32 on the surface of glioma cells and describe the design and characterization of a p32-specific CAR T that efficiently kills glioma cells both in differentiated and stem-like states. Cancer cell plasticity has been proposed to be an alternative mechanism to promote cancer cell diversity and to contribute to intra-tumor heterogeneity[42]. Plasticity in solid tumors has been linked to the epithelial-to-mesenchymal transition[43], and in GBM it confers the ability to shift from a differentiated state to an undifferentiated or stem-like state[44,45]. We also reported another interesting scenario in cancer cell plasticity: the transdifferentiation of glioma cells to TDEC, demonstrating the complexity of this phenomenon in brain tumors[34]. We found p32 to be expressed on the surface of differentiated cells, GSCs, and TDEC, and our in vitro studies demonstrated the ability of p32 CAR T cells to specifically recognize and efficiently kill all these populations of glioma cells representing different phenotypic/ differentiation states. GSCs have been reported to be more resistant to conventional therapy such as chemotherapy and radiotherapy, and some even blame their resistance to be the source of recurrence in GBM patients. Besides, we have previously shown that GSC differentiation into TDEC is independent of VEGF and FGF, suggesting that TDECs involvement in tumor angiogenesis might be one of the resistance mechanisms against

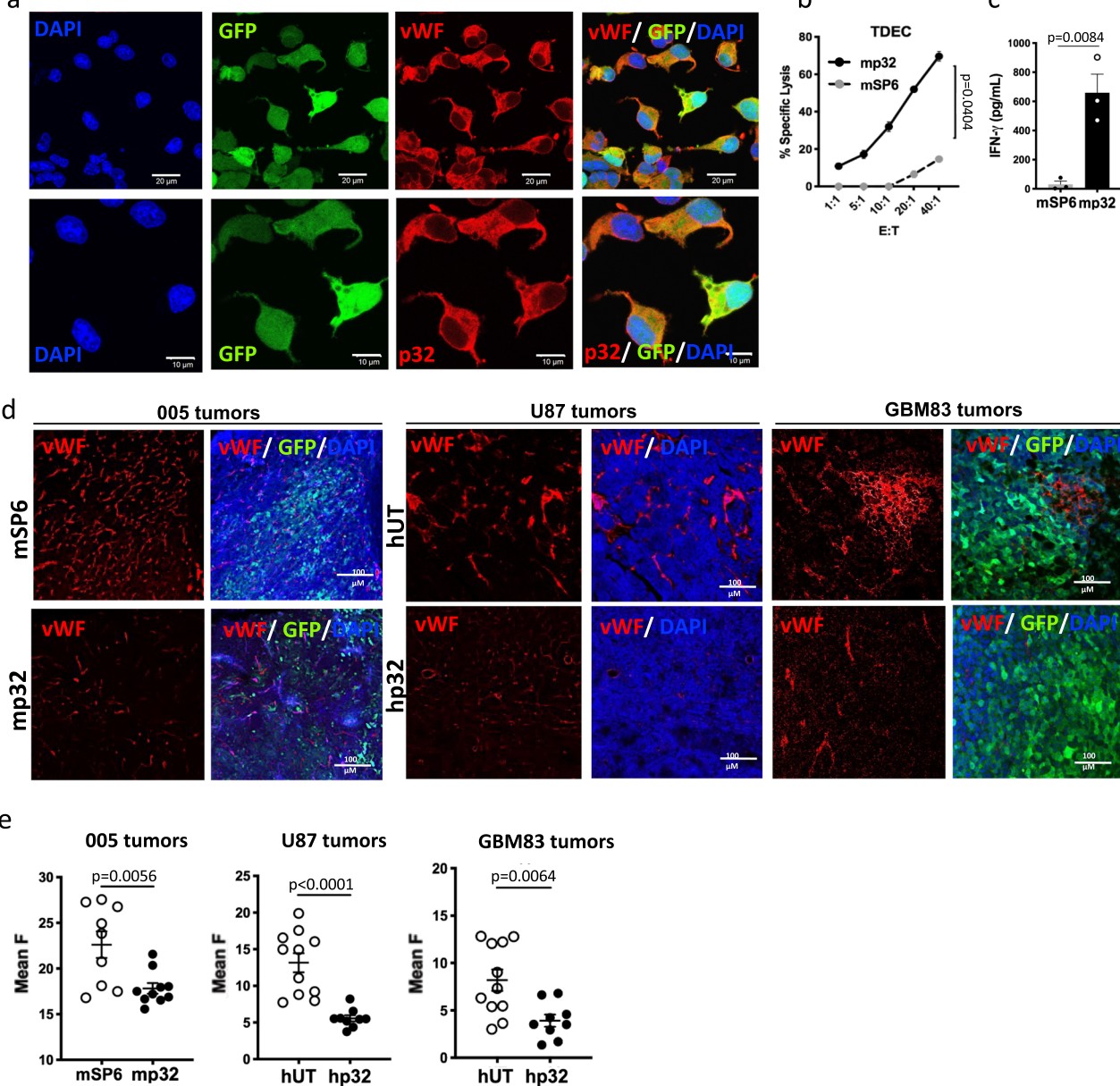

**Fig. 5 P32 CARTs exert an antiangiogenic action both in vitro and in vivo. a** 005 cells differentiated into tumor-derived endothelial cells (TDEC) in EGM medium with deferoxamide (DFO; to mimic hypoxia) were analyzed for Von Willebrand factor (vWF, endothelial marker) (upper panels) and p32 (lower panels) expression by confocal microscopy. Representative image of three independent experiments. **b** Cytotoxic action of mSP6 and mp32 (murine CAR) T cells against TDEC assessed by LDH cytotoxicity assay. Data represent mean ± SEM. $N = 3$ independent experiments. Unpaired t test with Welch's correction was used for statistical analysis. Two-tailed $P$ value is shown. **c** Recognition of TDEC by CAR T cells (measured by IFN-γ release) Data are shown as mean ± SEM, each dot is the average for an individual experiment ($N = 3$). **d, e** Tumor tissue stained for von Willebrand factor (vWF) and analyzed by confocal microscopy shows a clear decrease of blood vessels in the p32CAR treated animals. **e** Quantification of vWF staining. Data are shown as mean ± SEM. Each dot shown in the graph represents average of three measurements done per slide. A total of 9 (mSP6, hp32 for both models), 10 (mp32) or 11 (hUT) slides originated from 3 different mice were stained per group. For **c, e** unpaired t test was used and two-tailed $P$ value is shown. Source data are provided as a Source Data file.

anti-VEGF therapies. The ability of p32 CAR T cells to recognize and eliminate glioma cells in different phenotypic/differentiated states provides an advantage to this strategy over the existing conventional therapies and the possibility of combining these therapeutic approaches for the treatment of GBM.

In addition to the diversity within the tumor (intra-tumor heterogeneity), genetic profiling of GBM samples from different patients revealed inter-tumor heterogeneity, suggesting that at least three molecular subtypes of GBM exist: proneural (PN), classical (CL), and mesenchymal (MES)[33,46]. These molecular subtypes can also co-exist within the same tumor (spatial heterogeneity) or change over the course of tumor progression or as a result of therapy (temporal heterogeneity)[46–48]. Both our TCGA analysis and flow cytometry experiments using MES and PN GSCs showed that p32 expression is observed in all subtypes of GBM. Altogether, expression of p32 was found in both low and high-grade gliomas, different cell states, and different GBM subtypes, making it a glioma-relevant target for immunotherapy. But even with all these positive features that further support p32 as suitable TAAs in gliomas, and the significant overall survival

observed in our pre-clinical models, mice bearing GBM tumors eventually succumb to the disease. In all the in vivo experimental models, p32-CAR T cells induced the regression of GBM growth, and it is important to highlight that this effect was assessed on endogenous levels of p32 expression, no overexpression or ex vivo modifications were performed to the transplanted glioma cells. A positive indication of on-target effect is the decrease of antigen expression in the tumors treated with p32-CAR T cells compared to UT treated group, and no toxicity was observed following local and systemic injection of p32-specific CAR T cells. We believe that in solid tumors, due to the significant tumor heterogeneity, more than one tumor antigen should be targeted, using for example technologies such as the CART-BiTE recently reported in GBM[32], or the convertible CARs where multiple antigens can be simultaneously or sequentially targeted[49]. Here the benefit of p32 as a surface-expressed antigen in the tumor and not in healthy tissue adds one more option to the limited pool of CAR T targetable antigens in GBM. As nicely summarized in a recent review, one of the main challenges CAR T therapies for GBM faces is intra/inter-tumor heterogeneity and antigen loss[50]. The degree of expression of each antigen tested in the aforementioned CAR T GBM clinical trials presented regional, temporally and inter-patient heterogeneity. And while successful eradication of TAA positive cells was observed in each different study, it was also accompanied by the progression of glioma cells not expressing the relevant antigen. Increasing the pool of TAAs to target and the engineered CAR T repertoire in GBM will help overcome in part this particular obstacle. Furthermore, although we observed a short window of CAR specific activity (tumor growth by bioluminescence imaging) a multifactorial in situ immunosuppressive response drives the expression of exhaustion markers such us PD−1 on infiltrating T cells, leading to tumor re-growth. These observations suggest the possibility of combining CAR T adoptive transfer with inhibition of checkpoint players such as the PD1/PD-L1 axis.

In conclusion, our study provided evidence of specific surface expression of p32 in gliomas, and the potent antitumor and anti-angiogenic activity of p32-specific CAR T cells against glioma and tumor-associated and derived endothelial cells, indicating it may be of relevance for the treatment of glioma patients.

## Methods

**Materials**. DMEM, RPMI, fetal bovine serum (FBS), L-glutamine, penicillin, streptomycin, and mycoplasma detection kit were purchased from Biological Industries Ltd. (Kibbutz Beit HaEmek, Israel). DMEM/F-12 (1:1)w/L glut w/o HEPES(Ce) \500 ml (Cat. No. 11320074), N−2 Supplement (100X), Liquid \5 ml (Cat. No. 17502048), B−27 Supplement W/O Vit A (50X/10 ml) (Cat No. 12587010), B−27 Supplement (50X) (Cat. No. 17504044), Glutamax (Cat. No.10565-018), were purchased from Gibco. Recombinant Human FGF-basic (Cat No. 100-18B) was purchased from Peprotech. Recombinant Human EGF (Cat No. G5021) was purchased from Promega. RetroNectin (Cat No. T202) was purchased from Takara (Japan). Ficoll-Paque PLUS (Pharmacia Biotech, Uppsala, Sweden). GentleMACS C tubes for cell separation (Cat. No. 130-093-334), CD45 MicroBeads for cell isolation (Cat. No. 130-052-301) were purchased from Miltenyi Biotec (Bergisch Gladbach, Germany). IFN-γ mouse IFN-γ DuoSet ELISA (Cat No. DY485) and IFN-γ Human IFN-γ DuoSet ELISA (Cat No. DY285B) were purchased from R&D systems. Cell tracer violet (Cat No. C34557A) was purchase from Invitrogene. Percoll medium (Cat. No. p4937) and all other chemical reagents, including salts and solvents, were purchased from Sigma-Aldrich (Rehovot, Israel).

**Cell culture**. Human embryonic kidney 293T cells (HEK 293T), U87, U118, U178 were obtained from the American Type Culture Collection (ATCC, Manassas, VA, USA). U251 human GBM cell line was obtained from the European Collection of Authenticated Cell Cultures (ECACC) (Porton Down, Salisbury, UK). Cell lines from ATCC have been authenticated; ATCC uses morphology, karyotyping, and PCR-based approaches to confirm the identity of human cell lines. The ICLAC identifies U118 as misidentified line, a derivative of U138, possibly sharing a common donor. We only used this cell line once in this study to check the expression of p32 on surface (FACS analysis). U251 cell line was authenticated by the European Collection of Authenticated Cell Cultures (ECACC) using

morphology, karyotyping, PCR-based techniques, and Cytochrome oxidase I assay, following manufacturer validated procedures. GBM 83, GBM1005, GBM1027, GBM1079, GBM1051 were obtained from Prof. Ichiro Nakano. G179, G26, G7 were obtained from Prof. SM Pollard/Peter Dirks. BL line was obtained from Prof. Santosh Kesari. NCH421K line was obtained from Prof. C. Herold-Mende and received from Prof. Tambet Teesalu. GBM83 is a patient tumor-derived GSC (newly diagnosed female, 72 years old, 30–35% Ki67 index)[30]. Human primary cells H−6067, H−6034, H−6013, H−6044 were purchased from cell biologics, and N7805−100 was purchased from Gibco. 005 is a murine GSC line generated from a lentiviral HRasV12 induced tumor in a p53−/− knockout mouse[29] and AFFR53 are transformed primary astrocytes with HRas-shp53 lentivirus[51]. O1 cells were derived from FGFR1mut-shp53 lentiviral-induced tumor. Murine-derived cells lines (005, AFFR53, and O1) and patient-derived cell lines were not authenticated by the authors. For a complete list and summary of all the cell lines please refer to Supplementary Table 1.

HEK293T cells, PG13 (ATCC CRL-10686) human packaging cells, U87, U118, U178, U251, and AFFR53 cell lines were grown in Dulbecco's modified Eagle's medium (Biological Industries) supplemented with 10% FBS (Biological Industries), 2 mM glutamine (Biological Industries), 2 mM sodium pyruvate (Biological Industries), 100 units/ml penicillin, and 50 mg/ml streptomycin (complete DMEM media). GP + E86 (MPGE) (ATCC CRL-9642) murine packaging cell lines were grown in complete DMEM media supplemented with 10% high iron FCS (Defined Calf Serum, Iron Suppl. Tarom). 005 and O1 murine GSC were grown in DMEM F12 media (Gibco), supplemented with Glutamax 1:100 (Gibco),100 units/ml penicillin, and 50 mg/ml streptomycin, N2 supplement 1:100 (Gibco), 2.5 μg/ml Heparin (Sigma), 20 ng/ml FGF (Peprotech), 20 ng/ml EGF (Peprotech). GBM83 human glioma patient-derived cells were grown in DMEM F12 media (Gibco), supplemented with Glutamax 1:100 (Gibco),100 units/ml penicillin, and 50 mg/ml streptomycin, B27 supplement without Vitamin A 1:50 (Gibco), 2.5 μg/ml Heparin (Sigma), 20 ng/ml FGF (Peprotech), 20 ng/ml EGF (Peprotech). GBM1005, GBM1027, GBM1079, GBM1051, GBM1, BLGFP, NCH421K human glioma patient-derived cells were grown in DMEM F12 media (Gibco), supplemented with 2 mM glutamine (Biological Industries), 100 units/ml penicillin, and 50 mg/ml streptomycin, B27 supplement without Vitamin A 1:50 (Gibco), 2.5 μg/ml Heparin (Sigma), 20 ng/ml FGF (Peprotech), 20 ng/ml EGF (Peprotech). G179, G26, G7 were grown on laminin-coated plates (Sigma L2020-1mg/ml, dilute 1:100) in DMEM F12 media (Gibco), supplemented with Glutamax 1:100 (Gibco),100 units/ml penicillin, and 50 mg/ml streptomycin, 15 mM Hepes, 0.7% glucose, MEM NEAA (Gibco 11140 – 035, 1:100), 0.012% BSA (Gibco 15260-037), 50 μM 2-Mercaptoethanol, N2 supplement 1:100 (Gibco), B27 supplement with vitamin A 1:50 (Gibco 17504−044) 50 μg/ml Heparin (Sigma), 10 μg/ml FGF (Peprotech), 10 μg/ml EGF (Peprotech). Human primary cells were thawed and propagated according to the supplier's instructions. All cell lines were routinely tested for mycoplasma using the EZ-PCR-Mycoplasma test kit (Biological Industries). All cells used were tested negative for mycoplasma contamination. All cells were incubated at 37 °C in 5% $CO_2$.

**Gene knockdown and luciferase/GFP expression**. Either lentivector containing GFP-Luc[35] or Puro-Luc (Addgene Cat. No. 17477) was used to transduce AFFR53, U87, 005, and GBM83 cells. AFFR53 and 005 p32 knockdown (KD) cells were generated by lentiviral transduction. The short hairpin targeting mouse p32 (shp32): 5′-GGATGAGATTGGTCACGAAGA−3′ or an irrelevant short hairpin (shIrr): 5′-CTAACACTGGGTTATACAA−3′ were cloned into the NheI site of the p156RRLsin vector (a third-generation lentiviral vector)[13]. HEK293T cells were seeded on 100-mm plates and co-transfected with lentiviral vector plasmid and packaging plasmids. Viral supernatant was collected 48 h post transfection, concentrated by ultracentrifugation and added to the cells. Infected cells were selected using Puromycin (1 μg/mL).

**Generation of p32-specific CAR and Retrovirus production**. Genes encoding for the scFv-p32[25] were cloned into a pMSGV retroviral vector containing CD28 costimulatory domain followed by FcγR signaling domain[28]. We added a P2A-mcherry fluorescent reporter cassette for easy assessment of transduction efficiency and a FLAG tag at the scFv N-terminus for surface expression analysis by FACS.

HEK293T cells were co-transfected with CAR retroviral vectors and packaging vectors and the supernatant containing the retroviral particles was used to infect GP+E86 (ATCC CRL-9642), and PG13 (ATCC CRL 10686) cells to obtain stable packaging cell lines.

**Transduction of primary lymphocytes**. To generate p32 hCAR T cells, peripheral blood mononuclear cells (PBMCs) were separated on Lymphoprep (Ferenius Kabi Norge AS) centrifuge gradient, and activated on non-tissue culture 6-well plates pre-coated with anti-hCD3 (Cat. No. 317326, clone OKT3, Lot B317658, 1 μg/ml) and anti-hCD28 (Cat. No. 302934, clone CD28.2, Lot B318445, 1 μg/ml) antibodies from Biolegend (San Diego, California, USA) for 48 h. On day 3, activated T cells (2.5 × 10^6 cells/well) were transduced with retroviral supernatant obtained from CAR-PG13 cells using "spin-infection" on RetroNectin 6-well coated plates with the addition of 100 U/ml IL-2 (Cat. No. 200-02, PeproTech, USA). Human T cells from healthy donors were purchased from Magen David Adom (Blood services

Center) in Israel. All experiments complied with protocols that were approved by Institutional Review Board at Tel Aviv University and Magen David Adom.

In a similar fashion, to obtain p32 mCAR T cells, retroviral supernatant obtained from CAR-GPE86 packaging cell line was used to transduce activated murine splenocytes by "spin-infection" on RetroNectin plates ($5 \times 10^6$ cells/well). Splenocytes from syngeneic mice were activated with soluble anti-mCD3 (Cat. No. 100331, clone 145-2C11, Lot B266702, 30 ng/ml) and anti-mCD28 (Cat. No. 102112, clone 37.51, Lot B212837, 30 ng/ml) antibodies from Biolegend (San Diego, California, USA), and 100 U/mL of IL-2 for 2 days before transduction was performed. Transduced murine lymphocytes were expanded for 24 h in the presence of 350 U/mL IL−2.

Transduction efficiency was estimated by analyzing FLAG cell surface expression, as well as mcherry expression on retrovirally transduced cells and comparing to either isotype control stained or untransduced T cells (UT).

### CAR T-cell tumor recognition

*IFN-γ secretion.* Human ($0.5 \times 10^6$ cells) or murine ($1.2 \times 10^5$ cells) cancer cells were seeded in 24 wells plate, followed by addition of human ($1 \times 10^6$ cells) or murine ($2.5 \times 10^5$ cells) CAR T in RPMI complete media, respectively. Cell culture supernatants were harvested and assayed for interferon (IFN)-γ 18 h later using the mouse or human IFN-γ ELISA Kit (R&D Systems), according to the manufacturer's instructions.

*Cytotoxicity.* For the specific lysis assays, luciferase-expressing tumor targets ($1 \times 10^4$ cells) were co-cultured with varying amounts of p32 CAR T, Sp6 CAR T or UT T cells, for 18 h. Luminescence was measured using a IVIS lumina IIITM system (Perkin Elmer) immediately upon addition of Luciferin (D-Luciferin Firefly, potassium salt Cat No. LUCK250, GOLDBIO, Gold Biotechnology). Other cytotoxicity assays included LDH cytotoxicity assay (Sigma-Aldrich, Cat. No. 11644793001), and co-culture of GFP positive target cells with varying amounts of p32 CAR T, Sp6 CAR T or UT T cells for 3–5 days. Cells were collected, stained and analyzed by flow cytometry.

*Proliferation assay.* T cells were labeled with CellTrace™Violet probe (Molecular Probes™Thermo Fisher Scientific), according to manufacturer's instruction and co-cultured with glioma cells at an E:T ratio of 3:1. CellTrace™Violet dilution was analyzed gating on CAR T/T cells on day 3 using flow cytometry.

### Western blotting

To assess p32 expression, proteins were extracted from cultured cells following lysis in ice cold NP40 lysis buffer (50 mM Tris pH 8, 150 mM NaCl, 1% NP−40) supplemented with fresh proteases inhibitors (Invitrogen, USA). Cell lysates were loaded into a 10% acrylamide/bis-acrylamide gel and then transferred onto nitrocellulose filter and blocked with 5% skim milk in Tris-HCL buffer with 1% Tween. Mouse anti-GC1qR/p32 (Abcam, Cat. No. ab24733, clone 60.11, lot no. GR3208495-4 dilution 1:1000) and Rabbit anti-tubulin (Santa cruz, Cat. No. Sc53646, clone 10D8, lot no. K0515, dilution 1:1000) antibodies were incubated with the nitrocellulose filter overnight at 4 °C followed by incubation with goat anti-mouse-HRP (Jackson ImmunoResearch Cat. No. 115-035-166, Lot 124784, Dilution:1:10,000) or goat anti-rabbit HRP (Jackson ImmunoResearch Cat. No. 115-035-166, Lot 128223, Dilution:1:10,000) for 1 h at room temperature.

### Flow cytometry antibodies

Anti-mouse CD3ε-APC (Biolegend, Cat No. 100311, clone 145-2C11, lot B291091, dilution 1:100). Anti-mouse CD8a-alexa 488 (Biolegend, Cat No. 100723, clone 53-6.7, lot B254526, dilution 1:200). Anti-mouse CD4-BV 785 (Biolegend, Cat No. 100453, clone GK1.5, lot B236697, dilution 1:100). Anti-mouse CD25-PE (Biolegend, Cat No. 101904, clone 3C7, lot B247733, dilution 1:50). Anti-mouse LAG3-PerCP-Cy5.5 (Biolegend, Cat No. 125211, clone C9B7W, lot B261544, dilution 1:200). Anti-mouse PD-1-PerCP-Cy5.5 (Biolegend, Cat No. 135208, clone 29F.1A12, lot B302661, dilution 1:200). Anti-human CD3ε-APC (Biolegend, Cat. No. 300439, clone UCHT1, lot B278610, 5 μl per test). Anti-human CD4-FITC (Biolegend, Cat No. 317408, clone OKT4, lot B234953, 5 μl per test). Anti-human CD8a ef450 (Biolegend, Cat No. 48008842, clone RPA-TB, lot 1988089, 5 μl per test). Anti-human CD25-FITC (Biolegend, Cat No. 302604, clone BC-96, lot B253407, 5 μl per test). Anti human LAG3-APC (Biolegend, Cat No. 369212, clone 7H2C65, lot B294090, 5 μl per test). Anti human CD45RA-APC (Biolegend, Cat No. 304112, clone HI100, lot B276540, dilution 1:200). Anti human CCR7-FITC APC (Biolegend, Cat No. 353216, clone G043H7, lot B194490, dilution 1:200). Anti human PD-1-FITC (Biolegend, Cat No. 621611, clone A17188B, dilution 1:200). Rabbit anti-FLAG (Cell Signaling, Cat. No. 14793, clone 145-2C11, Lot B206702, Dilution 1:400). Rabbit IgG Isotype control (Cell Signaling, Cat. No.3900, Dilution: same concentration as a corresponding primary antibody). Anti-mouse/human C1QBP-PE (Santa Cruz, Cat. No. sc-23884, Clone 60.11, Lot L0403, Dilution: 20 μl per test).

### Flow cytometry and immunofluorescence staining

Efficiency of T cell transduction was assessed using anti-FLAG antibody along with mcherry fluorescent reporter. For flow cytometry analysis 100,000 cells were stained with the appropriate antibodies according to the antibodies manufacture instructions. Briefly, the cells were incubated with TruStain FcX™-Fc blocker CD16/32 (Biolegend, Cat.

No. 101320, clone 93, Lot no. B275367, dilution 1:100) for 15 min to reduce non-specific staining, followed by staining with primary antibodies listed above for 30 min at 4 °C. Cells were washed and resuspended in PBS. The intracellular staining for IFN-γ was performed with PE-conjugated antibody (Biolegend, Cat. No. 505807, Clone XMG1.2, Lot no. B240650, dilution 1:100) and for p32 with C1QBP-PE (Santa Cruz, Cat. No. sc-23884, Clone 60.11, Lot L0403, 20 μl per test) following fixation and permeabilization of cells using BD Cytofix/Cytoperm kit (BD Bioscience, 554714) according to the manufacturer's protocol. Fluorescence intensity was assessed using an Attune NxT Flow Cytometer and analysis was performed using Kaluza 2.1 software (Beckman Coulter, USA).

Negative controls included isotype antibodies and where appropriate, untransduced T cells stained with the test antibodies. Gating strategy for all flow cytometry experiments is included in Supplementary Fig. 9.

For confocal fluorescence imaging analysis, mice were perfused with 1× PBS and fixed with 4% paraformaldehyde. Brains were collected and coronal sections (30–40 μm) were cut using a HM450 Microtome (ThermoFisher Scientific). Floating sections were incubated overnight at 4 °C with the following antibodies: rabbit anti-p32 polyclonal antibody (gift from Prof. Tambet Teesalu, stock 4.5 mg/ml, Dilution: 1:100), rabbit anti-vWF (Abcam, Cat. No. ab6994, Lot GR3180938-1, Dilution 1:100), followed by goat anti-rabbit-AlexaFluor488 (Abcam, Cat No. ab150077, dilution 1:500), or goat anti-rabbit Cy3 (Jackson ImmunoResearch, Cat No 111-165-144, Lot 123834, dilution 1:400). Nuclei were counterstained with DAPI (Molecular Probes) at 1 μg/ml. Images were obtained using a Zeiss LSM 800 Confocal Microscope and ZEN 2.3 software. Images were analyzed using Fiji/ImageJ 2.0.0.

### Histology

Mice were anesthetized and transcardially perfused with cold PBS, followed by 4% PFA. Organs were collected, further fixed by incubation in fresh 4% PFA overnight, and subsequently embedded in paraffin. Slides were prepared and stained with Hematoxylin and Eosin (Sigma-Aldrich) and photographs were taken using a Nikon Eclipse Ti microscope.

### Ethics

Fresh surgical samples of gliomas were obtained from Tartu University Clinics, Tartu, Estonia under protocols approved by the Ethics Committee of the University of Tartu, Estonia (permit #243/T27). All methods for the use of human samples were carried out in accordance with relevant guidelines and regulations. Informed consent was obtained from all patients.

### Animals and ethics statement

All mice were housed in the Tel Aviv University animal facility under specific pathogen-free conditions. Groups of up to five mice per IVC cage (Lab-Products) were housed on a 12 h light/dark cycle, on autoclaved ASPEN wood chips bedding, at an ambient temperature of 22 °C ± 1 °C, with humidity controlled at 50%, had ad libitum access to regular laboratory chow (Altromin1324; Altromin, Lage, Germany), and were provided with UV-irradiated and micro-filtered Hydropac system for water. Animal experiments were approved by the animal care and use committee (IACUC) of Tel Aviv University (approval protocol no. 04-16-073) and conducted in accordance with NIH guidelines.

### Animal studies and bioluminescence imaging

C57Bl/6J and Nude mice were purchased from Envigo Jerusalem Israel. Male and female 8- to 12-week-old mice were used in all experiments. 005 ($3 \times 10^5$), U87-GFP ($3 \times 10^5$) or GBM83-Luc ($3 \times 10^4$) cells were stereotaxically injected into the hippocampus (AP = −2.0, ML = −1.5, DV = 2.3) of C57BL/6 J mice or Nude mice. C57BL/6 J mice were preconditioned with an intraperitoneally (i.p.) injection of 200 mg/kg cyclophosphamide (Sigma-Aldrich, Israel) 24 h prior to CAR T cells administration. CAR T transduced or untransduced control T cells were systemically ($1 \times 10^7$ cells/mouse), intratumorally ($1 \times 10^7$ cells/mouse) or intraventricularly ($2.5 \times 10^6$ cells/mouse) (AP = 0.3, ML = −2.0, DV = 3.0) injected to brain tumor-bearing mice. To monitor tumor growth, mice were i.p. injected with 150 mg/kg D-luciferin (BioGold) and imaged using the IVIS system (Xenogen) 10 min after injection. The bioluminescence images were analyzed using Living Image software 4.7.3 (Caliper Life Sciences).

### Tissue processing

Brain tumors (GFP + tissue resected under fluorescent microscope) were dissociated using a Neural dissociation kit (Miltenyi Biotec) according to the manufacturer's instructions and the resulting cell suspension was cleaned of debris (myelin) via Percoll (Sigma) density gradient centrifugation. CD45- cells were enriched using anti-mouse-CD45 magnetic-microbead and MS columns (Miltenyi Biotec) according to the manufacturer's instructions.

For dissociation of lung tissue, after organ extraction, tissue was cut in small pieces and then incubated with collagenase type IV (Worthington #LS004186), Dispase II (Sigma-Aldrich, 4942078001) and DNAse I (Sigma-Aldrich, 11284932001) in a 37 °C water bath for 40 min under constant stirring. The digested mixture was then strained and red blood cells were lysed prior to staining.

### Statistics

Biological replicates of at least 3–4 mice/donors were compiled for each in vitro experiment and two mice/donors for the in vivo experiments, unless otherwise specified in figure legends. Experiments were repeated at least 2–3 times independently with similar results. Student's *t* test, 1 and 2-way ANOVA tests statistical analysis and graphing were performed using GraphPad Prism 8 software

for MAC. Data represent mean ± SEM, n values are listed in figure legends. Log-rank (Mantel-Cox test) analysis was used to determine the statistical significance of Kaplan–Meier survival plots. $P < 0.05$ was considered statistically significant.

**Reporting summary**. Further information on research design is available in the Nature Research Reporting Summary linked to this article.

## Data availability

The glioma patient sequencing data analyzed in this study are available and were downloaded from http://gliovis.bioinfo.cnio.es (Rembrandt adult dataset was selected for tumors from different histological types and GBM molecular subtypes, and CGGA adult dataset to compare primary and recurrent tumors). The remaining data generated or analyzed during this study are available within the article and its supplementary information files. Source data are provided with this paper.

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

## Acknowledgements
We thank Dr. Moran Rawet Slobodkin from the laboratory of Prof. Zelig Eshhar, Immunology research center, Tel Aviv Sourasky Medical Center (TASMC) for her help in generating packaging cell lines. This work was supported by the FP7 Marie Curie Carrier Integration Grant, the Israel Cancer Research Fund (Research Career Development Award), the Israel Cancer Association, the Cancer Biology Research Center (CBRC), and the Israel Science Foundation (#1310/15).

## Author contributions
D.F.M. designed and supervised the project. L.R.N., I.M., and S.T. performed most of the experiments. T.W., A.G.L., and Z.E. provided valuable reagents, helped with human T cell transduction, and reviewed the data. L.A.V. provided the p32 scFv and reviewed the data. M.H. and T.T. performed the p32 staining of human GBM samples. L.R.N., I.M., and D.F.M. wrote or edited the manuscript. All authors have seen, corrected, and approved the final manuscript.

## Competing interests
The authors declare no competing interests.
