## [Peer Review File · Nature Communications]

Reviewers' comments:

Reviewer #2 (Remarks to the Author):

[Editorial note: reviewer #2 did not provide any comments to the Authors, but he/she expressed his/her remarks directly to the Editors, as reported in the main body of the decision letter]

Reviewer #3 (Remarks to the Author):

This is an interesting study evaluating CAR T cells targeting p32, a potentially new and attractive target for GBM. While this is a novel target and some suggestive findings presented in this manuscript, the rigor of experiments and areas of overstatement of results limit enthusiasm for publication in Nature Communications.

- More context and background for p32 as a TAA as well as significance/rigor of prior studies in the introduction would better set the stage for the experiments presented.

- It does not seem appropriate (although I defer to Nature Communications editorial staff) to re-publish data as part of supplemental figures (Sup Fig 1, Sup Fig 2). Instead this should be referenced and/or repeated as original data in this manuscript.

- Figure 1 does not support claims in results. Authors need to more accurately reference figures.
 - o Statement that p32 expression is increased in all GBM subtypes is not clear from Figure 1b. This figure seems to indicate that the classical subtype has higher levels of p32 than other subtypes. Please be more specific and provide statistics between each subtypes and non-tumor control as in A.

- o There is no clear trend showing that p32 is associated with a worst prognosis for GBM (Fig 1C) or for the mesenchymal subtype (Fig 1F). The survival curves show a no significant difference (ns) between p32 high vs low.

- The analysis of expression of p32 in human GBM samples needs to be expanded. Please provide IHC/IF of patient resection samples to evaluate p32 staining on normal brain and a panel of tumor samples. Also, addressing cell surface vs intracellular expression of p32, is important.

- Include FACS for AFR53 and GBM 005 p32-KD lines to show specificity of flow cytometry staining.

- Figure 2 should include data to show transduction efficiency for multiple experiments/donors as well as T cell phenotypes (CD4/CD8 ratios, memory markers and exhaustion markers) for the ex vivo expanded cells. Showing one representative experiment is not adequate.

- Figure 3 does not show proliferation of CAR T cells as stated in the Results (line 123). Flow plots represent a percentage and as tumor cells are killed it would be expected that the % of CAR-Ts increases. To show proliferation total T cells need to be enumerated or a CFSE dilution assay performed. Also, please enumerate CAR+ cells, not just CD3.

- In Figure 4, it is surprising the weak killing of both GBM83 and U87 (see panels b and d), especially since persistence of T cells are detected. Are CAR+ cells persisting post tumor challenge—Fig 4 only shows CD3 vs CAR? Please show CAR T cells levels post tumor challenge. Are low antigen expressing targets not being killed in these assays? Please show p32 levels of surviving tumor cells. This is important in light of Fig 5h showing low p32 tumor cells not being targeted in vivo by p32-CAR T cells.

- For Figure 5, ffLuc differences between mice are not convincing, and it is unclear if this is a reproducible result. It is also unclear what is plotted (median vs mean), what error bars represent and what statistical test was done. Please also show all mice in Fig 5e for these time points. Also, please be clear if this difference is repeated in more than one experiment.

- More extensive evaluation for the safety for p32 CAR T cells is important. Authors could evaluate whether CAR T cells show evidence of activation when injected IV by evaluating activation markers from CAR T cells isolated from lung tissue and blood, as well as evaluate normal brain following IC delivery into the parenchyma. ffLuc-engineered CAR T cells would establish if p32 CARs are activated or proliferate

- The antiangiogenic effects of p32-CARs is not adequately addressed. This is a significant conclusion of the paper (see Title) but not sufficiently established by the studies presented.

- o Showing the trans-differentiation of the 005-GSC line is not sufficient to establish the differential p32 expression on tumor-associated endothelial cells and the specificity of CAR-T targeting. A more in-depth analysis is needed to demonstrate the specific tumor endothelial targeting by p32 CAR-Ts.

- o Importantly, any tumor-targeted CAR would be expected to reduce tumor vasculature by

decreasing tumor burden (through direct cytotoxicity) and secretion of inflammatory cytokines such as IFN. It is very likely the observed decrease in vWF in the tumor microenvironment is non-specific and would occur with any GBM-targeted CAR. If the tumors were engineered to be responsive to SP6-CARs, would a specific decrease in tumor vascular be observed?

- Throughout the manuscript insufficient information is provided for statistical tests used. It is important to specify what exact statistical test was used for comparison. In some panels it is unclear whether means or medians are plotted (example Fig 4d) and whether error bars are SD or SE.

- Insufficient information is provided for how many times experiments have been repeated and how many donors have been used for human CAR studies. It seems that most comparisons are within an individual assay, and it is not clear how robust findings are across donors and experiments.

Below we provide point-by-point responses (Queries in black, response to Reviewers in blue):

Referee #1:

1. CAR construct is different from the usual construct; their CAR has CD28 co-stimulation domain fused to FcR-gamma instead of 4-1BB- ζ

Years ago, when we started the project we had the 2nd generation CAR vector with CD28-gamma (from Prof. Zelig Eshhar, my PhD mentor), we thought that as proof-of-concept, this construct was suitable. Today there is not a clear advantage over each of the constructs and no clinical trial that compares all the possible combinations side by side. In the Her-2 clinical trial in GBM, the costimulatory molecule used is CD28 as in our construct.

In a follow up study we can definitely fine-tune our CAR construct (as many other investigators in the field of CAR T did) and find the best performer, using mouse models of course.

2. P32 is overexpressed in all 3 subtypes of GBM, but more prominently in classical and mesenchymal GBM.

Yes, that is correct, based on mRNA levels of expression, p32 is prominently expressed in classical and mesenchymal GBM, with non significant differences between these two subtypes. Probably these two subtypes will benefit more from p32-specific CAR T therapy, although we believe that positivity for p32 expression (IHC staining and not mRNA expression) could be a better criteria for patient selection.

We do not understand what is the major concern of the reviewer here, but we have clarified the mRNA expression of p32 better in our results section.

3. Target antigen has low expression on normal cells (Fig 1A and 1B).

Based on previous Fig1A and Fig1B results (now moved to Supplementary Fig2a and 2b), mRNA levels and **NOT** target antigen, has low expression in normal tissue, and still the differences are statistically significant.

But we would like to remind the reviewer that for CAR T therapy, the levels of expression of the protein are the most important and even more so, the expression of the antigen on the surface of the tumor cells. To further address this important point, we have now used primary cortical astrocytes and fibroblast to show that although p32 is expressed intra-cytoplasmic in these cells, NO expression is detected on the surface of these cells (new Fig 1d)

In Fig 1b it is clear that p32 is hardly expressed in the non-tumor tissue adjacent to the tumor lesion. In the revised version of our manuscript we have added representative images and quantification of normal human brain tissue stained with p32 antibody to further support our results.

One additional point to remind the reviewer is that in normal cells, p32 is primarily expressed in mitochondria and only in tumors it was found expressed on the surface (line 74-77 in original manuscript; 86-89 in revised manuscript version).

4. In vitro testing done in mice as well as human cell lines. These data are supportive of authors' conclusion.

We follow the exact work flow as in every other CAR T study. We identified our TAA, we designed the CAR construct and YES, the first thing we did is to validate the functionality of

our p32-CAR construct in vitro using both our murine and human cell lines. The results obtained support our hypothesis, that p32 expressed on the surface of glioma cells is a good target for CAR T therapy.

Unfortunately, the reviewer has not suggested any other experiment we could have conducted other than the ones we did in our study, so we cannot alleviate his/her concerns.

5. The in vivo testing done in syngeneic, orthotopic xenograft and PDX mice models.

That is correct, we indeed performed our in vivo validation in three different models: a) syngeneic, b) human cell line in xenograft, c) patient derived cell xenograft.

I can cite here numerous studies that only used one of the abovementioned models to validate their novel CAR functionality in vivo. We actually took an extra effort to show it in all three different models.

Here I would like to highlight the advantage we had in our model system to be able to use a syngeneic model. Many studies cannot even show that.

An additional and important comment here, is that many studies overexpress their TAA in glioma cells to validate their CAR T in vivo/in vitro. In our study, all our models express their parental endogenous levels of p32, they were not “transfected/transduced” with any exogenous p32 vector to overexpress the target.

6. The mice data are not convincing; for example, in description they mentioned 8 mice per group but in BLI graphics showed only 3 mice.

In the revised version of the manuscript we have added more representative images of the mice and in the quantification next to the images included more mice (Fig 4d-e). The experiment is representative of two independent experiments performed with two different donors.

7. For the syngeneic model they infused two doses of T cells at 10 million CART/dose at 2 different time points, but the mice survival is not dramatic.

First of all, we were probably not clear enough in our original manuscript, but we always reported the TOTAL amount of T cells infused to the mice. Since our efficiency of transduction is on average 30%, not all the infused cells are CAR T positive. We thank the reviewer and in our revised version we have clarified this very important detail.

Unfortunately, there is no single syngeneic model system that shows “dramatic” survival and not many studies show the results using syngeneic models in GBM.

I think the important conclusion from this experiment, which we have emphasized in our revised version, is that CAR T were able to reach the tumors when administered systemically and even have an effect on the tumor compared to the control treated group.

Just for the sake of comparison, below are the results of CD70-CAR T in a syngeneic model using the exact same dose of 1×10^7 cells/dose at 2 different time points (actually they infused more CAR T+ cells, since their efficiency of transduction was ~60% and ours was 30%).

[REDACTED]

Fig. taken from Jin L. et al, Neuro-Oncology 2017.

8. For xenografts they injected 30000 tumor cells intracranially on day 0 followed by 10 million CART cells intracranial and 25 million cells intravenous on day 3 which is a dose that is 10 times higher than we normally use. But the mice survival does not correlate with the higher dose of T cells.

We are sorry to have confused the reviewer, our mistake here. Please notice that in the Results section in the text (line 159-160 in original article) and in the methods (line 341-342, original article) we report the actual dose we use in that specific experiment: 1×10^7 intratumor and **2.5×10^6 intraventricularly**. We did not use intravenous route in our xenograft models. The dose reported in the legend of that figure is a typo mistake (2.5×10^7), we have now fixed it of course.

We are using a very aggressive GBM patient derived xenograft, which latency is only on average 20 days. That is the reason why we inject the CAR T cells on day 3 (we of course verified that all the mice have lesions by BLI).

For the sake of the reviewer's comparison with his/her system/dose, if 35×10^6 was 10 times the dose, that means the reviewer is using a dose of approx. 3.5×10^6

We are injecting a dose of 12.5×10^6 total cells, **4×10^6 CAR T positive cells** (calculated based on 30% average efficiency of transduction) which is then a very similar dose to that of the reviewer.

9. The only novelty of this paper is they tested CART cells against an antigen previously never tested.

We can reply by listing a long number of publications where they ONLY novelty was to report a CAR T against an antigen previously never tested.

1. Jin, L. et al, Neuro-Oncology 2017
2. Du H, et al. Cancer Cell 2019
3. Morgan, RA, et al, Rosenberg SA, Hum Gene Ther 2012
4. Johnson LA, et al. Sci Transl Med, 2015 (they used same TAA as 3, novelty: humanized scFV)
5. Mount, CW et al. Nat Med 2018. TAA is not novel, but they target a different pediatric tumor (DIPG).

A simple google search will extend the list to many more.

Referee #2:

[They have commented confidentially to the Editor that the heterogeneous expression of p32 in tumors, in their opinion, makes it unsuitable as a target for glioblastoma.]

As the referee knows, one of the hallmarks of glioblastoma is heterogeneity. We never claimed that p32 was expressed in all the cells in the tumor, on the contrary, we are aware of the heterogeneity of expression and not only of our newly identified TAA for CAR T therapy, but also of any other TAA identified so far for the treatment of solid tumors with CAR T cells. In addition, we never claimed that p32 was exclusively expressed on brain tumors (lines 76-77 in original manuscript), which actually can be an advantage, in terms of having a CAR T that can be used for the treatment of multiple types of tumors. Her2, for example, is not exclusively expressed in gliomas and is currently in clinical trials for multiple tumor types; same is true for TAA reported in the literature: B7-H3 (Du H, et al, Cancer Cell, 2019), CD70 (Jin L. et al, Nat Commun 2019), and we can go on with the list.

Just to put the heterogeneity of p32 in context, we will compare its heterogenous expression with the other three targets that are today in clinical trials for GBM: IL13Ra2, Her2 and EGFRviii.

1. IL13Ra2 (data from clinical trial reported in *Brown CB, et al, NEJM 2016*)

[REDACTED]

From the text: "H score of 100 (with no staining in 30% of cells, weak-intensity staining in 30%, moderate-intensity staining in 20%, and high-intensity staining in 10%) (Fig. S1 in the Supplementary Appendix, available with the full text of this article at NEJM.org). The H score, a method of quantitating immunohistochemical results, is based on the following formula: (3×the percentage of strongly staining cells)+(2×the percentage of moderately staining cells)+(1×the percentage of weakly staining cells), resulting in a range of 0 to 300."

2. Her2 (data from clinical trial reported in *Ahmed N, et al. JAMA Oncol 2017*)

[REDACTED]

Based on the table, 82% of the patient had 1-50% expression, while 35% of the patients enrolled to the study had 1-25% expression of Her2.

3. EGFRviii (data from clinical trial reported in *O'Rourke DM, Sci Trans Med 2017*)

[REDACTED]

Prof. Santosh Kesari, reported in his study using an array of 100 brain tumors (Fogal V. *Oncotarget* 2014), that median high expression of p32 is 20%. It is clear that not all GBM patients express p32, but from the ones that do express p32, the clinical trial cut could be set at 10-15% high intensity (or calculate the H score as in the IL13R2a clinical trial). The calculation of the H score from Kesari's array is 150.

We also found data from the Human protein atlas, where p32 protein is expressed in 50% of GBM patients and from 12 patients analyzed, 8 showed high to medium expression.

[REDACTED]

Referee #3:

“This is an interesting study evaluating CAR T cells targeting p32, a potentially new and attractive target for GBM. While this is a novel target and some suggestive findings presented in this manuscript, the rigor of experiments and areas of overstatement of results limit enthusiasm for publication in Nature Communications.”

1. More context and background for p32 as a TAA as well as significance/rigor of prior studies in the introduction would better set the stage for the experiments presented.

We thank the reviewer for this suggestion and we have already added a new paragraph in the introduction with more background on p32.

2. It does not seem appropriate (although I defer to Nature Communications editorial staff) to re-publish data as part of supplemental figures (Sup Fig 1, Sup Fig 2). Instead this should be referenced and/or repeated as original data in this manuscript.

We have now included in our revised version our own staining of patient derived samples and quantification of expression (Fig 1a)

3. Figure 1 does not support claims in results. Authors need to more accurately reference figures.

o Statement that p32 expression is increased in all GBM subtypes is not clear from Figure 1b. This figure seems to indicate that the classical subtype has higher levels of p32 than other subtypes. Please be more specific and provide statistics between each subtype and non-tumor control as in A.

We follow the reviewer's suggestion and in the revised version we have sent all the mRNA analysis of p32 expression to Supplementary Fig 2. We have added the non-tumor controls in all the graphs and show the statistical analysis for each group. All three GBM subtypes significantly expressed p32 compared to non-tumor samples with significant difference of expression between classical and proneural subtypes.

We would like to remind the reviewer that although these data is part of the characterization and significance of p32 expression in brain tumors, and we thought it was important to present it, it is less relevant as a target for CAR T immunotherapy. Maybe our mistake here was to be tempted to follow the same format as other studies that reported CAR T in gliomas, to show the expression of mRNA levels of the TAA in their first figure.

o There is no clear trend showing that p32 is associated with a worst prognosis for GBM (Fig 1C) or for the mesenchymal subtype (Fig 1F). The survival curves show a no significant difference (ns) between p32 high vs low.

We agree with the reviewer that we did a poor job presenting the characterization of p32 and its expression in gliomas and in GBM in particular. The graph in Fig 1C (original version) actually showed that there is no significant difference between GBM patients with IDH1wt or IDH1mut. And this is actually in correlation with the analysis of expression of p32 in those groups, both IDH1wt and IDH1mut GBM express similar levels of p32 mRNA.

And indeed, in the classical subtype, that expresses higher levels of p32, high levels of p32 are associated with worst prognosis. In the revised manuscript version we decided to show the overall survival of glioma patients expressing low vs. high p32 to avoid confusion (Supplementary Fig 2d).

- The analysis of expression of p32 in human GBM samples needs to be expanded. Please provide IHC/IF of patient resection samples to evaluate p32 staining on normal brain and a panel of tumor samples. Also, addressing cell surface vs intracellular expression of p32, is important.

Following the reviewer's suggestion, we have now added in the revised version representative images of IF section of patients and normal brain tissue evaluated for p32 staining and quantification graph of the samples (new Fig 1a).

From tumor sections it is very difficult to address the differences between surface and intracellular expression. However, we performed this comparison taking normal cells: primary cortical astrocytes and fibroblast and our glioma cells and analyzed the intracytoplasmic and surface expression of p32 by FACS (new Fig 1d).

- Include FACS for AFR53 and GBM 005 p32-KD lines to show specificity of flow cytometry staining.

We have included the flow cytometry of AFR53 and 005 p32KD performed by flow cytometry staining if Supplementary Fig 3 c-d.

4. Figure 2 should include data to show transduction efficiency for multiple experiments/donors as well as T cell phenotypes (CD4/CD8 ratios, memory markers and exhaustion markers) for the ex vivo expanded cells. Showing one representative experiment is not adequate.

In the revised version we have now included the phenotypic characterization of our CAR T in four different "mouse donors" and four independent human donors. All the analysis was included in Fig 2.

5. Figure 3 does not show proliferation of CAR T cells as stated in the Results (line 123). Flow plots represent a percentage and as tumor cells are killed it would be expected that the % of CAR-Ts increases. To show proliferation total T cells need to be enumerated or a CFSE dilution assay performed. Also, please enumerate CAR+ cells, not just CD3.

The flow plots actually do represent the expansion of total T cells, since we know the number of cells we plated initially in each well, and we collected/acquired data from fixed volume in the FACS, we can infer number of cells remaining in each well. But we agree with the reviewer that to show proliferation a better assay is to follow dilution of labeled T cells. So in the revised version of the manuscript we have labeled the T cells with CellTracer-Violet and assessed proliferation of CAR T+ cells by gating for this population in the FACS analysis (new Fig 3b and 3d).

6. In Figure 4, it is surprising the weak killing of both GBM83 and U87 (see panels b and d), especially since persistence of T cells are detected. Are CAR+ cells persisting post tumor challenge—Fig 4 only shows CD3 vs CAR? Please show CAR T cells levels post tumor challenge. Are low antigen expressing targets not being killed in these assays? Please show p32 levels of surviving tumor cells. This is important in light of Fig 5h showing low p32 tumor cells not being targeted in vivo by p32-CAR T cells.

This is a fair and very interesting point raised by the reviewer. However, the E:T ratio in this assay was (1:5), meaning less effector cells since we intended to follow the expansion of the effector population and not much the killing in this specific assay. Also the incubation time was much longer compared to the cytotoxic assay performed in previous panels a-c. The sensitivity of these two instruments are different and based on different readouts.

In the revised version we separated the killing assay (new Fig 3a and 3e) from the proliferation and expansion of the CAR T population (Fig 3b and 3d and Supplementary Fig 4, respectively).

7. For Figure 5, ffLuc differences between mice are not convincing, and it is unclear if this is a reproducible result. It is also unclear what is plotted (median vs mean), what error bars represent and what statistical test was done. Please also show all mice in Fig 5e for these time points. Also, please be clear if this difference is repeated in more than one experiment.

In the revised version we have added more representative images of BLI as well as a graph summarizing the radiance (see exact units description in Fig 4) for all the mice in the experiment. All the experiments were performed twice and in the case of human CAR T with two independent donors. We have added all this information both in the legends of the figures as well as in the Methods-Statistics section.

8. More extensive evaluation for the safety for p32 CAR T cells is important. Authors could evaluate whether CAR T cells show evidence of activation when injected IV by evaluating activation markers from CAR T cells isolated from lung tissue and blood, as well as evaluate normal brain following IC delivery into the parenchyma. ffLuc-engineered CAR T cells would establish if p32 CARs are activated or proliferate.

We appreciate the reviewer's suggestion. We have now extended our evaluation of the safety of p32-CAR T cells. In the syngeneic model we injected i.v. labeled engineered T cells (CellTracer) and dissociated the lung tissues as described in Materials & Methods section. We gated for CD3+ T cells and CAR T+ cells and compared the proliferation of control (mSP6) to p32-specific CAR T (mp32). We also compared the status of activation of these cells by FACS. In addition we collected tissue from different organs for pathology analysis (H&E) and overall we observed no differences between the two groups. For the human CAR T models we infused healthy mice with the same dose of either hUT or hp32 engineered T cells and pathology analysis revealed no differences. All results are shown in Supplementary Fig 5.

9. The antiangiogenic effects of p32-CARs is not adequately addressed. This is a significant conclusion of the paper (see Title) but not sufficiently established by the studies presented.

- o Showing the trans-differentiation of the 005-GSC line is not sufficient to establish the differential p32 expression on tumor-associated endothelial cells and the specificity of CAR-T targeting. A more in-depth analysis is needed to demonstrate the specific tumor endothelial targeting by p32 CAR-Ts.

We understand the concerns of the reviewer and we appreciate the comments.

We have repeated the experiment but this time in addition to confocal microscopy analysis we used FACS analysis of dissociated tumors to evaluate the percentage of endothelial CD31+ cells (see Supplementary Fig 7). Again we observed a decreased in the percentage of endothelial cells in the p32-CART treated group compared to control group.

- o Importantly, any tumor-targeted CAR would be expected to reduce tumor vasculature by decreasing tumor burden (through direct cytotoxicity) and secretion of inflammatory cytokines

such as IFN. It is very likely the observed decrease in vWF in the tumor microenvironment is non-specific and would occur with any GBM-targeted CAR. If the tumors were engineered to be responsive to SP6-CARs, would a specific decrease in tumor vascular be observed?

This is indeed a very interesting point, that can only be addressed experimentally in our system. To address this very important point we engineered glioma cells to overexpress ErbB2 (GBM83-ErbB2, see Supplementary Fig 7a-b). We next validated the in vitro functionality of an ErbB2-specific CAR T (hN29, Globerson-Levin et al. Molecular Therapy 2014) against the engineered GBM83-ErbB2 target cells (Supplementary Fig 7c-e). Finally, we treated GBM83-ErbB2 tumor bearing mice with either hUT or hN29 CAR T and followed tumor growth and survival (Supplementary Fig 7f-h). And for the most relevant part of the experiment, at end point, as we did before with the p32-CART system, we analyzed tumor sections for vWF staining and dissociated tumors for CD31+ endothelial expression by FACS. As shown in Supplementary Fig 7i-k, no significant differences were observed between the groups.

Although tumor burden was reduced at some point during the treatment in both model systems (either targeting p32 or ErbB2), at end point when we analyzed the tumors, ONLY in the p32-CAR T treated group we observed the anti-angiogenic effect. There are several explanations on how tumors can develop in the absence of “conventional” blood vessels (e.g. vasculogenic mimicry) but these experiments are out of the scope of this paper.

10. Throughout the manuscript insufficient information is provided for statistical tests used. It is important to specify what exact statistical test was used for comparison. In some panels it is unclear whether means or medians are plotted (example Fig 4d) and whether error bars are SD or SE.

We agree with the reviewer. In the revised version we have added all the information both in the legends and in Methods—Statistics section.

11. Insufficient information is provided for how many times experiments have been repeated and how many donors have been used for human CAR studies. It seems that most comparisons are within an individual assay, and it is not clear how robust findings are across donors and experiments.

We have now included all the information requested by the reviewer in the revised version of the manuscript.

REVIEWER COMMENTS

Reviewer #3 (Remarks to the Author):

Results: line 90, states that Figure 2 includes both TCGA and Rembrandt databases, but figure legend only references Rembrandt. Also, please clarify the datasets for each panel evaluated in Sup Fig 2. Is this restricted to grade III and IV malignant gliomas? what is the n for each glioma group. Assuming low and high is based on median expression level?

Results: line 125, the engineering of murine CAR T cells is unclear. Are both murine and human T cells engineered with the human CAR? Or have the authors generated a murine equivalent to the CAR shown in Fig 2A (i.e. murine CD28 etc). Also, for a general reader the description of murine p32/glioma lines vs human p32/glioma lines is not well described. Upfront authors should clarify the ability of this antibody to bind both murine and human p32 and identify 005/AAR53 as murine gliomas in the results. Also, are there any data to support that the 2.15 clone has comparable affinity for murine vs human p32.

Results: line 130, is the intent of Figs 2f-2i to show that this CAR does not tonically signal? Please consider giving providing the conclusions from these experiments.

Figure 4f, g h: what is the time point for tumor harvest ? At endpoint? Date ranges?

Minor points:

Abstract: line 38, suggest editing to among "one of" the most important

Abstract: line 43, suggest adding concluding sentence for perspective.

Results: line 88, possible word missing "although it is primarily expressed...."

Evidence that the antibody x-reacts with mouse for safety

- was the CAR the same for human and mouse T cells? Does the CAR x-react with murine p32?

Please clarify cytokines used for expansion of mouse T cells and what CD3/CD28 antibodies used for expansion.

Reviewer #4 (Remarks to the Author): to replace Reviewer #1

In this manuscript, the authors identify p32 as a novel tumor associated antigen target expressed on the surface of glioma cells. They proceed to generate p32 CAR T cells and found them to recognize and specifically eliminate murine and human p32 expressing glioma cells and tumor derived endothelial cells in vitro, and to control tumor growth in orthotopic syngeneic and xenograft mouse models. The authors take a very comprehensive approach to the development and validation of their CAR T cells, in particular in their rigorous in vivo validation of CAR T cell functionality using 3 different mouse models: syngeneic, human cell lines in xenograft, and patient-derived cell xenograft. Experimental methodology is solid and in keeping with published literature, and findings are significant and of interest to the glioma community. Some concerns with overinterpretation of data and overstatement of conclusions can be addressed with toning down some statements in the manuscript with respect to the suitability and context of p32 as a therapeutic target in glioma. In addition, the concerns listed below should be addressed prior to publication in Nature Communications.

Major points:

1. The authors are presenting data in Figure 1C and 1D to show exclusive or increased expression of p32 in glioma models. Although the authors present data showing no p32 expression on normal murine astrocytes and fibroblasts, no comparable data is shown on normal human cells using flow cytometry. Considering the differences between human and murine biology, the authors have not

unequivocally answered whether p32 expression is “restricted to tumor cells” (line 111-112) and cannot make the claim that “p32 [is] a novel tumor-associated antigen in low- and high-grade gliomas” (line 112-113).

2. In Figure 1C and 1D, flow cytometry data is presented as histograms. It is not clear what percentage of cells express p32. Density plots for all flow cytometry analysis with calculated percentages should be provided in supplementary data.

3. Heterogeneous expression of p32 is a viable concern for pursuing a CAR T cell therapy in gliomas. However, there are very few TAAs in the glioma literature that show uniform cell surface expression in all glioma cells (for example, EGFRv3 is only expressed on 30-60% of cells within a classical subtype GBM, and has been avidly pursued as a therapeutic and CAR T cell target in gliomas). Furthermore, the rationale of targeting glioma stem cells is that although they represent a minority fraction of the tumor cell population, they generate all other cells within the tumor hierarchy and thus represent a tractable target. Immense intratumoral and intertumoral heterogeneity in gliomas may explain differences in p32 expression and the authors show that a large amount of patient-derived glioma stem cell models express p32 using flow cytometry (Figure 1C). Flow cytometric characterization of p32 in additional patient-derived GSC models and normal human cells would strengthen the arguments that p32 is a TAA in glioma and there is a significant cohort of gliomas with high p32 expression.

4. The authors have well addressed the differential expression of p32 in tumor cells at the surface and in normal cells in the mitochondria as a factor in support of pursuing it as a therapeutic target for immunotherapy. However, it would be valuable for authors to provide some speculation as to why P32 is being trafficked to the cell surface in glioma when it's typically a mitochondrial matrix protein.

Minor points:

1. In Supplementary Figure 1, the authors present immunohistochemical analysis of p32 expression in cancerous and normal tissue. However, these data were originally published in Fogal V. et al, *Cancer Res* 2008, 68(17): 7210–7218, as indicated in the figure legend, and cannot be republished as original data in a new manuscript. A reference to the original article is sufficient.
2. In Supplementary Figure 2E, the authors “validated p32 protein expression by western blot analysis in both murine and human GBM [glioblastoma] cells” (line 100-101). However, only one of these lines are classified as Grade IV glioblastoma (U118) with high confidence. The authors indicate that 005 and AFR53 are mouse glioma models, GBM83 is a human glioma stem cell model, and U87 model is not confidently diagnosed as Grade IV glioblastoma by ATCC. In fact, long term culture of U87MG cells have led to drastic changes in their DNA profile by STR profiling (PMID 27582061). The authors should re-write this part of their manuscript with greater accuracy.
3. What contributions of tumour reduction were specific to cyclophosphamide?
4. Why use lung tissue used for collection and analysis of CAR T proliferation and activation?
5. Is it intraventricular injection? They very clearly say intravenous tail vein injection elsewhere in the paper. If intraventricular, how are authors confirming intraventricular delivery during surgery?
4. Line 61: Minimal? Not minimal
5. Line 66: Make clear that there has been no efficacy of these CAR-Ts in clinical trials
6. Line 119: Any particular reason for selecting an scFv that targets the C1q domain of P32?
7. Line 151: IFN
8. Line 203: Cannot be sure if it is antigen escape or elimination or both. PD-1 is also a marker of activation alone, could be exhaustion but better to characterize in combination with other markers such as LAG-3 as done earlier in the paper
9. Line 209/210: Have to be careful in implicating the immunosuppressive microenvironment given that some in vivo models are immunodeficient mice that lack a TIME.
10. TDEC marker to confirm transdifferentiation? Please specify.
11. Line 237: Why not co-stain by IHC for P32 and CD31 to confirm P32 expression on tumor-associated endothelial cells?

Sheila Singh

Below we provide point-by-point responses (Queries in black, response to Reviewers in blue):

Referee #3

Results: line 90, states that Figure 2 includes both TCGA and Rembrandt databases, but figure legend only references Rembrandt. Also, please clarify the datasets for each panel evaluated in Sup Fig 2. Is this restricted to grade III and IV malignant gliomas? what is the n for each glioma group. Assuming low and high is based on median expression level?

We apologize for our mistake. We have now corrected the sentence and indicated in the legend of the Fig which dataset was used for each evaluation. We also added a table below each graph to indicate the number of patients for each group (Supplementary Fig1). The analysis we performed includes both low and high grade gliomas. In Suppl. Fig1d low and high refers to median expression levels of p32, we have included the information in the legend.

Results: line 125, the engineering of murine CAR T cells is unclear. Are both murine and human T cells engineered with the human CAR? Or have the authors generated a murine equivalent to the CAR shown in Fig 2A (i.e. murine CD28 etc). Also, for a general reader the description of murine p32/glioma lines vs human p32/glioma lines is not well described. Upfront authors should clarify the ability of this antibody to bind both murine and human p32 and identify 005/AAR53 as murine gliomas in the results. Also, are there any data to support that the 2.15 clone has comparable affinity for murine vs human p32.

Thanks for the reviewer's comment. We have now better clarified all the points raised by the reviewer in the revised manuscript:

1. Both the antibody and derived scFv recognize murine and human p32 as was previously described by our collaborator Luis Alvarez-Vallina (Sanchez-Marin D, JBC 2011). P32 is a highly conserved protein and the epitope recognized by the anti-p32 scFv shares 90% homology between mouse and human. Unfortunately, we do not have affinity data comparing murine vs. human p32, but for the reviewer perusal we attach below the results of an ELISA using purified 2.15 clone and either human or mouse p32 protein. These results show that the scFv in our CAR construct recognize both human and mouse p32 with similar capacity.

Both our data in the present manuscript and our collaborator's previous results show recognition of p32 by the scFv 2.15 clone/Antibody in both species.

2. We used a single CAR design for all our experiments with either murine or human glioma targets. The CAR design shares the same intracytoplasmic signaling domains: CD28-gamma. We have previously shown that although the CD28-gamma construct is from human origin, it is capable of signaling and activating both human and murine T cells:
Friedmann-Morvinski D et al. Blood 2005
Globerson-Levin A et al. Mol Ther 2014
Globerson-Levin A et al. Cancer Immunol Immunother 2020
3. We have added a supplementary table describing the murine and human p32 positive glioma cells used in the manuscript. Materials and methods section describes the human and murine glioma cells used for the *in vitro* and *in vivo* studies. And we have better clarified the origin of these cells in the text. In addition, in the "Functional evaluation of p32 CAR T cells *in vitro*" section we separated the murine system from the human system in two different paragraphs. I hope that all together, this helps the reader distinguish between the two models.

Results: line 130, is the intent of Figs 2f-2i to show that this CAR does not tonically signal? Please consider giving providing the conclusions from these experiments.

Yes, we appreciate the reviewer's comment and we have better clarify the intention of the experiments in the text and pointing to the fact that the p32-specific CAR does not tonically signal when transduced into T cells.

Figure 4f, g h: what is the time point for tumor harvest ? At endpoint? Date ranges?

Yes, the tumors analyzed in Figs 4f, g, and h were harvested at endpoint (we have added this information in the revised manuscript). For the group treated with hUT cells tumors were collected from days 15-22 post-tumor injection, and for the hp32 treated group from days 22-30 (See Fig 4c).

Minor points:

Abstract: line 38, suggest editing to among "one of" the most important.
We have edited the sentence to include "among one of"...

Abstract: line 43, suggest adding concluding sentence for perspective.
We had to slightly modify the abstract (and still remain in the 150 word limit) and have now included the concluding sentence "P32 CAR T cells could represent a novel therapeutic option for glioblastoma patients".

Results: line 88, possible word missing "although it is primarily expressed...."
We corrected the mistake, the word "it" was indeed missing.

Evidence that the antibody x-reacts with mouse for safety

• was the CAR the same for human and mouse T cells? Does the CAR x-react with murine p32?
We have addressed this point in the previous paragraph (2nd major point). The answer is yes, the same p32-CAR reacts with both murine and human p32. We have clarify this important point in the manuscript text.

Please clarify cytokines used for expansion of mouse T cells an what CD3/CD28 antibodies used for expansion.

We have added the missing information in the materials and methods section.

Reviewer #4

In this manuscript, the authors identify p32 as a novel tumor associated antigen target expressed on the surface of glioma cells. They proceed to generate p32 CAR T cells and found them to recognize and specifically eliminate murine and human p32 expressing glioma cells and tumor derived endothelial cells in vitro, and to control tumor growth in orthotopic syngeneic and xenograft mouse models. The authors take a very comprehensive approach to the development and validation of their CAR T cells, in particular in their rigorous in vivo validation of CAR T cell functionality using 3 different mouse models: syngeneic, human cell lines in xenograft, and patient-derived cell xenograft. Experimental methodology is solid and in keeping with published literature, and findings are significant and of interest to the glioma community. Some concerns with overinterpretation of data and overstatement of conclusions can be addressed with toning down some statements in the manuscript with respect to the suitability and context of p32 as a therapeutic target in glioma. In addition, the concerns listed below should be addressed prior to publication in Nature Communications.

We appreciate the reviewer's comments, we have rephrase some of the results' conclusions to avoid overstatements, and we address below the major and minor concerns raised by the reviewer.

Major points:

1. The authors are presenting data in Figure 1C and 1D to show exclusive or increased expression of p32 in glioma models. Although the authors present data showing no p32 expression on normal murine astrocytes and fibroblasts, no comparable data is shown on normal human cells using flow cytometry. Considering the differences between human and murine biology, the authors have not unequivocally answered whether p32 expression is "restricted to tumor cells" (line 111-112) and cannot make the claim that "p32 [is] a novel tumor-associated antigen in low- and high-grade gliomas" (line 112-113).

We appreciate the reviewer's comment, this is a very important point. We have changed Fig 1d to present comparable data showing now FACS analysis with murine and human GBM cells compared to mouse and human normal astrocytes and fibroblasts. In addition to the analysis of these two primary human cell lines we added in supplementary material four additional human primary cells: lymphocytes, kidney epithelial cells, lung fibroblasts and liver epithelium (Supplementary Fig 3). All these primary human cells showed no expression of p32 on surface, which further support our rationale to re-direct p32-CAR T cells for the treatment of GBM.

2. In Figure 1C and 1D, flow cytometry data is presented as histograms. It is not clear what percentage of cells express p32. Density plots for all flow cytometry analysis with calculated percentages should be provided in supplementary data.

In the revised version of the manuscript we have added new supplementary Figs 2 showing the density plots for each histogram showing surface expression of p32 and their corresponding calculated percentages.

3. Heterogeneous expression of p32 is a viable concern for pursuing a CAR T cell therapy in gliomas. However, there are very few TAAs in the glioma literature that show uniform cell surface expression in all glioma cells (for example, EGFRv3 is only expressed on 30-60% of cells within a classical subtype GBM, and has been avidly pursued as a therapeutic and CAR T cell target in gliomas). Furthermore, the rationale of targeting glioma stem cells is that although they represent a minority fraction of the tumor cell population, they generate all other cells within the tumor hierarchy and thus represent a tractable target. Immense intratumoral and intertumoral heterogeneity in gliomas may explain differences in p32 expression and the authors show that a large amount of patient-derived glioma stem cell models express p32 using flow cytometry (Figure 1C). Flow cytometric characterization of p32 in additional patient-derived GSC models and normal human cells would strengthen the arguments that p32 is a TAA in glioma and there is a significant cohort of gliomas with high p32 expression.

In the revised version of the manuscript in addition to the five patient derived GSC we presented previously in Figure 1C we added six more patient derived GSC in Supplementary Fig 2b. We are also presenting flow cytometry analysis of six normal human primary cells: astrocytes and skin fibroblasts (main Fig 1d), lymphocytes, renal epithelial cells, lung fibroblasts and liver epithelium (Supplementary Fig 3). Altogether, we believe this additional FACS analysis further support the argument that p32 is a TAA in glioma.

4. The authors have well addressed the differential expression of p32 in tumor cells at the surface and in normal cells in the mitochondria as a factor in support of pursuing it as a

therapeutic target for immunotherapy. However, it would be valuable for authors to provide some speculation as to why P32 is being trafficked to the cell surface in glioma when it's typically a mitochondrial matrix protein.

This point is indeed very interesting and although there are reports on the biological function of mitochondrial p32 in gliomas (cited in our manuscript), why is being trafficked to the cell surface remains unknown. In the revised version of the manuscript we have added a few sentences in the discussion with speculation on why this might be happening in gliomas.

p32 is also known as hyaluronic acid binding protein (HABP) (Deb TB, Datta K J Biol Chem. 1996). Hyaluronic acid (HA) is an important component of the brain extracellular matrix (ECM) and is one of the major components of GBM-ECM. In addition to anatomical and physical aspects, specific ECM components in GBM, such as Tenascin C (Angle I et al Oncogene 2020), fibronectin, and hyaluronan were shown to contribute in different aspects of tumor biology (e.g. invasiveness, proliferation, angiogenesis). Based on a recently reported study (DeSimone AM et al. Science Adv 2019), we speculate that increased levels of HA may disrupt the normal localization of p32 in the mitochondria and induce its translocation to the cell surface.

Differential expression of p32/HABP was also reported during progression of epidermal carcinoma, where overexpression of p32 in proliferating cells was correlated with high levels of HA. We speculate that surface expressed p32 interacts with HA and this interaction contributes to GBM highly invasive and proliferative capacity.

We have added a paragraph in the discussion section.

Minor points:

1. In Supplementary Figure 1, the authors present immunohistochemical analysis of p32 expression in cancerous and normal tissue. However, these data were originally published in Fogal V. et al, Cancer Res 2008, 68(17): 7210–7218, as indicated in the figure legend, and cannot be republished as original data in a new manuscript. A reference to the original article is sufficient.

We have removed this supplementary figure and cited the original paper in the manuscript.

2. In Supplementary Figure 2E, the authors “validated p32 protein expression by western blot analysis in both murine and human GBM [glioblastoma] cells” (line 100-101). However, only one of these lines are classified as Grade IV glioblastoma (U118) with high confidence. The authors indicate that 005 and AFR53 are mouse glioma models, GBM83 is a human glioma stem cell model, and U87 model is not confidently diagnosed as Grade IV glioblastoma by ATCC. In fact, long term culture of U87MG cells have led to drastic changes in their DNA profile by STR profiling (PMID 27582061). The authors should re-write this part of their manuscript with greater accuracy.

We have clarify the glioma cells used in now Supplementary Figure 1e and we have added in the revised version of the manuscript a new Supplementary Table 2 summarizing all the glioma cells used in the study.

3. What contributions of tumour reduction were specific to cyclophosphamide?

We used pre-conditioning with cyclophosphamide in our 005 syngeneic model and we did not observe any effect of cyclophosphamide treatment (single dose i.p. 200 mg/kg) on tumor growth/latency as shown in the graph attached below (for the reviewer perusal; n=8).

4. Why use lung tissue used for collection and analysis of CAR T proliferation and activation?

It was suggested by one of the reviewers that since tail vein injections will hit the lungs among the first organs (after the heart of course) assessing the activation state of the recovered CAR T cells will give additional evidence of their safety (related to “on-target off-tumor toxicity”).

5. Is it intraventricular injection? They very clearly say intravenous tail vein injection elsewhere in the paper. If intraventricular, how are authors confirming intraventricular delivery during surgery?

For clarification we used different routes of CAR T administration in the study. In our syngeneic 005 GBM model we administered the engineered T cells intravenously (i.v.). In the U87 xenograft model the CAR T cells were injected intratumorally and only in the patient derived GBM model we decided to add the intraventricular route based on the reported positive results in clinical trials (NEJM ref). Based on the Allen Mouse Brain Atlas we used the coordinates described in Materials and methods to inject intraventricularly. We have no means to confirm delivery during surgery.

4. Line 61: Minimal? Not minimal

Definitely a mistake, we have now used the words “potential and dangerous” instead.

5. Line 66: Make clear that there has been no efficacy of these CAR-Ts in clinical trials.

We have added to the sentence “limited antitumor response has been observed in these clinical trials” to make it clear that there is no efficacy reported yet. We also end the sentence mentioning that eventually the patients succumb to the disease.

6. Line 119: Any particular reason for selecting an scFv that targets the C1q domain of P32?

The honest answer is no, we obtained this specific scFv from our collaborator, Luis Alvarez-Vallina, who identified it using a scFv library and characterized its binding to p32.

7. Line 151: IFN

Thanks, we corrected the mistake.

8. Line 203: Cannot be sure if it is antigen escape or elimination or both. PD-1 is also a marker of activation alone, could be exhaustion but better to characterize in combination with other markers such as LAG-3 as done earlier in the paper.

The reviewer is correct, we cannot rule out any of the possibilities. Technically it was difficult to recover the engineered T cells from the tumors at end point and since we had limited material to work with we decided to stain for PD-1. At the time we didn't have different fluorophore combinations to check an additional marker and we picked from all the options PD-1 because most of the studies using checkpoint inhibitors usually block the PD-1/PD-L1 axis. In addition, we have evidence from our syngeneic GBM model of high expression of PD-L1 in the TME (CD45+ cells, see below) as well as the tumor fraction and I supposed we were biased to check this marker.

9. Line 209/210: Have to be careful in implicating the immunosuppressive microenvironment given that some in vivo models are immunodeficient mice that lack a TIME.

We agree with the reviewer and we have re-phrased this sentence. We can only speculate in regards to the immunosuppressive TIME when using these models and we can only cite what others have observed in patient treated with CAR T and evaluated the TIME (O'Rourke DM, STM 2017). We do have preliminary evidence supporting an immunosuppressive TIME in our syngeneic murine GBM model (see example below: tumor was dissociated and PD-L1 expression was assessed in CD45+ tumor infiltrating fraction).

10. TDEC marker to confirm transdifferentiation? Please specify.

The characterization and differentiation/transdifferentiation of tumor cells to tumor derived endothelial cells was previously reported in Soda Y. et al. PNAS 2011 (Fig 5 in that paper). In our study we used the same 005 GSC from the PNAS paper that were cultured in EGM-2 media + DFO (deferrioxamine, to mimic hypoxia) and confirmation was assessed by expression of GFP (reporter in tumor cells) and vWF (marker of endothelial cells). Please refer to Fig 5A in our manuscript.

11. Line 237: Why not co-stain by IHC for P32 and CD31 to confirm P32 expression on tumor-associated endothelial cells?

We tried to co-stain the floating sections we had left from this experiment (we cut with a microtome) with the antibodies we used for flow cytometry but technically we couldn't get any staining. We even tried different antigen retrieval protocols but we were not successful in achieving good quality staining (we got very high background).

Sincerely,

Dinorah Friedmann-Morvinski

REVIEWERS' COMMENTS

Reviewer #3 (Remarks to the Author):

The authors have adequately addressed all comments and I thank the authors for their thoughtful responses.

Reviewer #4 (Remarks to the Author):

The author has addressed all of our original concerns very well in this revised manuscript. We appreciate the additional characterization of 6 further GSC lines and the attempted in vivo experiment to address cyclophosphamide treatment, as well as attempted IHC. We only have a few minor points to address before our recommendation to publish this manuscript.

1. The authors mention that they attempted to address toxicity by extraction and assessment of lung tissue. However, if they are attempting to address "on-target, off-tumour" effects, then this tissue should express P32 at the cell surface, which is not suggested as the manuscript indicates intracellular expression in healthy tissues. Any toxicity from IV treatment should thus not be termed as an assessment of "on-target, off-tumour" effects, and the manuscript should highlight that off-target effects are not observed as the activation state is unchanged in healthy tissues such as the lungs.

2. We ask the authors to address inconsistencies in Figure and supplementary figure labelling between the manuscript text and actual figures.

Sheila Singh

Below we provide point-by-point responses (Queries in black, response to Reviewers in blue):

Reviewer #4

The author has addressed all of our original concerns very well in this revised manuscript. We appreciate the additional characterization of 6 further GSC lines and the attempted in vivo experiment to address cyclophosphamide treatment, as well as attempted IHC. We only have a few minor points to address before our recommendation to publish this manuscript.

1. The authors mention that they attempted to address toxicity by extraction and assessment of lung tissue. However, if they are attempting to address "on-target, off-tumour" effects, then this tissue should express P32 at the cell surface, which is not suggested as the manuscript indicates intracellular expression in healthy tissues. Any toxicity from IV treatment should thus not be termed as an assessment of "on-target, off-tumour" effects, and the manuscript should highlight that off-target effects are not observed as the activation state is unchanged in healthy tissues such as the lungs.

We appreciate the reviewer's comment, overall we attempted to address toxicity and safety of the administration of our p32-specific CAR T systemically (in the syngeneic immunocompetent GBM mouse model). Indeed, no expression was found on the surface of healthy tissue and we have now completed the sentence in the text following the reviewers suggestion: "Once again, no difference was observed between the irrelevant CAR and p32-specific CAR T cells (Supplementary Fig. 6c), indicating that off-target effect is not observed as the activation state of the injected CAR T cells is unchanged in healthy tissues such as the lungs."

2. We ask the authors to address inconsistencies in Figure and supplementary figure labelling between the manuscript text and actual figures.

We apologize for the mistakes, we have corrected all the inconsistencies between main figures and Suppl. Figures labeling both in the text and in the actual figure (e.g. labeling of panel in Fig 3h).

We would like to thank both reviewers and the editorial team for helping us improve the quality of our study with their comments/suggestions/corrections, and especially to you for your sustained interest in our work.

Sincerely,

Dinorah Friedmann-Morvinski